# FLOW OF REASONING: TRAINING LLMS FOR DIVERGENT PROBLEM SOLVING WITH MINIMAL EXAMPLES

## ABSTRACT

The ability to generate diverse solutions to a given problem is a hallmark of human creativity. This divergent reasoning is also crucial for machines, enhancing their robustness and enabling them to assist humans in many applications such as scientific discovery. However, existing approaches to multi-step reasoning with large language models (LLMs) have mostly focused only on reasoning accuracy, without further discovering more diverse valid solutions. For example, supervised fine-tuning can improve LLM reasoning quality, but requires extensive supervised data to capture the full range of possible solutions. Reinforcement learning aims to find limited highest-reward solutions while neglecting the solution diversity. To fill this gap, we propose Flow of Reasoning (**FoR**), an efficient diversity-seeking LLM finetuning method aimed at improving reasoning quality and diversity with minimal data. FoR formulates multi-step LLM reasoning as a Markovian *flow* on a DAG-structured reasoning graph. This formulation allows us to incorporate and adapt principled GFlowNet approaches, for finetuning LLMs to sample diverse reasoning paths with probabilities *proportional* to the (unnormalized) reward of target problems. Extensive experiments show that, with limited training examples (e.g., 15 examples), FoR enables the discovery of diverse, creative, high-quality solutions, greatly outperforming a wide range of existing inference and training methods across five challenging puzzle-solving tasks, including BlocksWorld (embodied reasoning), Game24 (math puzzle solving), Rubik's Cube (spatial reasoning), 1D-ARC (abstraction reasoning), and PrOntoQA (logical reasoning).

## 1 INTRODUCTION

Divergent problem solving is the ability to generate multiple diverse solutions to a given problem (Runco, 1991; Runco & Acar, 2012). As a hallmark of human intelligence, this ability drives creativity by uncovering novel ways to accomplish a task, providing more possibilities and adaptivity in different complex situations. Similarly, by encouraging machines to explore diverse solutions rather than confining to one reasoning path, we not only enhance machines' robustness (e.g., by ranking or aggregating different solutions) (Wang et al., 2022), but also empower automated systems that assist humans in generating ideas and thinking out-of-the-box, thereby potentially facilitating task completion (Shinn et al., 2024), education (Li et al., 2023a), and scientific discovery (Jain et al., 2023a).

State-of-the-art reasoning with large language models (LLMs), however, has largely focused on improving only the problem-solving *accuracy* with the topmost solution, without moving a step further to discover more *diverse* valid solutions. Specifically, *inference* methods, such as CoT (chain of thought, Wei et al., 2022), ToT (Yao et al., 2024), RAP (Hao et al., 2023), and others (Chen et al., 2024b; Besta et al., 2024), rely heavily on the underlying pretrained LLM's capability and decoding algorithms to obtain diverse reasoning solutions. Moreover, the search-based inference (Yao et al., 2024; Hao et al., 2023; Chen et al., 2024b; Besta et al., 2024) can be computationally costly when searching for multiple reasoning paths. On the other hand, *finetuning* methods improve the inherent abilities of the underlying LLMs. However, the popular supervised finetuning (SFT) (Yue et al., 2023; Yu et al., 2023c) often demands extensive supervision data to capture the full diversity of solutions, which can be costly to label in many applications. Alternatively, reinforcement learning (RL), such as proximal policy optimization (PPO, Schulman et al., 2017), trains LLMs to generate

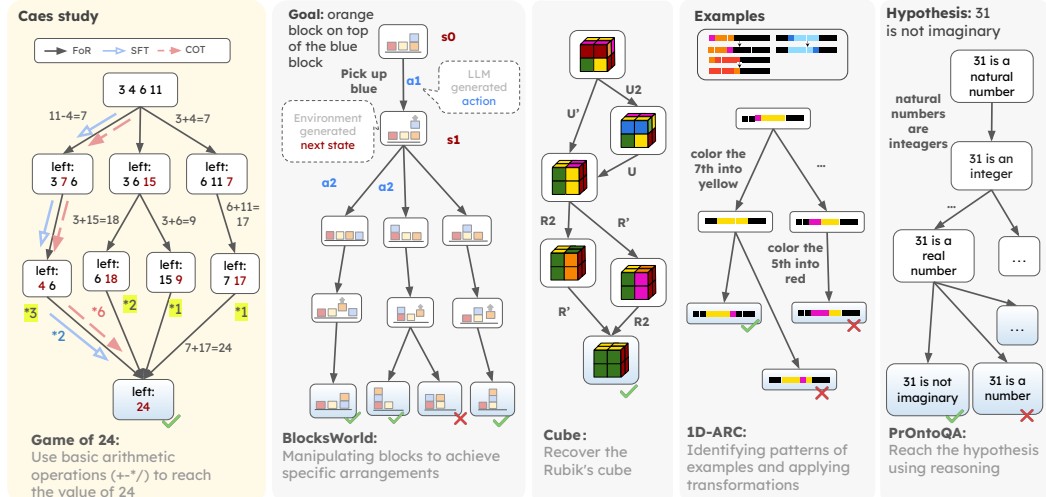

Figure 1: Multi-step LLM reasoning as a Markovian flow on five tasks, forming DAG-structured reasoning graphs. In the example of Game24 (left), we sample 20 reasoning paths from each method, respectively. Baseline methods such as SFT and CoT generate only one valid solution (leftmost path) repetitively (e.g., SFT generates this solution twice out of the 20 attempts), while our method FoR discovers three additional unique solutions.

the highest-reward reasoning solution and overlooks solution diversity. As shown in the case study in Figure 1, limited solutions are found by the above-mentioned methods.

To overcome the limitations, we introduce Flow of Reasoning (**FoR**), a data-efficient approach that finetunes LLMs for diverse reasoning with only minimal data. FoR draws inspirations from generative flow networks (GFlowNets) for amortized diverse sampling (Bengio et al., 2021) that have been studied in different domains like molecule synthesis (Koziarski et al., 2024; Kim et al., 2024a) and operation scheduling (Zhang et al., 2023a). In particular, FoR enables diversity-seeking finetuning of multi-step LLM reasoning, to sample high-quality reasoning paths with probabilities *proportional* to the reward of target problems (as opposed to reward *maximization* in conventional RL). To this end, we formulate multi-step LLM reasoning from a Markovian flow perspective (Figure 2), where each reasoning step corresponds to an edge (action) that leads to the next node (state) in a flow graph. The reasoning process thus forms a flow that travels step-by-step from an initial state to the terminal states of the target problem. Based on this new formulation, we introduce the trajectory balance objective and adapt efficient exploration methods from the recent GFlowNet studies, enabling effective finetuning of LLMs to align with the task reward using only 15 input examples.

FoR differs crucially from the recent GFlowNet applications on autoregressive sequence generation with or without LLMs (Hu et al., 2023a; Malkin et al., 2022a). In particular, contrary to the token-level modeling in the previous work, FoR introduces higher-level modeling at the granularity of reasoning steps. This combines the best of the GFlowNet sequence generation (Hu et al., 2023a; Malkin et al., 2022a) and the aforementioned search-based LLM reasoning (Yao et al., 2024; Hao et al., 2023) while overcoming their limitations, by enabling more flexible DAG-structured reasoning graphs, more efficient handling of complex multi-step reasoning problems, and thereby greatly improved reasoning quality and diversity as shown in §4. We evaluate the divergent problem-solving capability of the proposed approach on five puzzle-solving problems that have proven challenging for LLM reasoning, including *BlocksWorld* that involves embodied reasoning (Kambhampati et al., 2024), *Game24* involving math puzzle solving (Yao et al., 2024), *Rubik's Cube* involving spatial reasoning (Ding et al., 2023), *1D-ARC* involving abstraction reasoning (Xu et al., 2023b), and *PrOntoQA* involving logical reasoning (Saparov & He, 2022). Empirical results show that FoR, with limited (e.g. about 15) training examples, generates diverse, high-quality solutions, greatly outperforming a wide range of baselines with 20% - 85% improvements, including supervised training methods like SFT, reward-maximizing reinforcement learning like PPO, diversity-seeking approaches GFN-CoT and various decoding methods, and advanced inference methods like CoT, ToT, GoT, and RAP. Ablation studies further validate the key designs in FoR that lead to robustness and effectiveness.

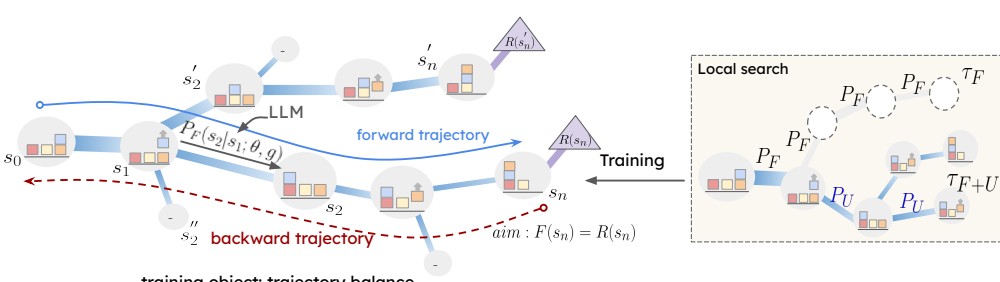

training object: trajectory balance

Figure 2: **Left:** The forward policy $P_F(s_t|s_{t-1}; \theta, g)$ in the flow-based formulation is parameterized as LLM and finetuned with the trajectory balance objective (Eq.5) to achieve the desired flow $F(s_n) = R(s_n)$ on all terminal states $s_n$. **Right:** FOR incorporates local search with a destroy-and-reconstruction process to augment informative trajectories in training (§3.2.2). This facilitates efficient exploration and improves policy learning.

## 2 RELATED WORK

**LLM reasoning.** Recent LLMs (Achiam et al., 2023; Touvron et al., 2023; Chowdhery et al., 2023) have shown strong potential in tackling complex reasoning tasks (Hu et al., 2023c; Zhang et al., 2023d; Yu et al., 2023b). **(1) Fine-tuning LLMs**, including supervised fine-tuning (SFT) and reinforcement learning (RL), is a key method for improving LLM reasoning abilities. **SFT**, leveraging large and high-quality datasets of reasoning chains, has proven highly effective (Yu et al., 2023c; Yue et al., 2023; Yuan et al., 2024a). **RL** techniques like PPO are widely used for optimizing reward-driven behavior in LLMs (Ouyang et al., 2022; Bai et al., 2022; Havrilla et al., 2024). However, both approaches tend to limit solution diversity. **(2) Prompting-based methods** engages LLMs in a step-by-step thinking process. Chain-of-Thought (CoT) (Wei et al., 2022) enhances LLM performance by guiding them through intermediate steps to reach the final answer. Building on CoT, methods like ToT (Yao et al., 2024) and GoT (Besta et al., 2024) model reasoning as tree and graph searches, enabling exploration of multiple paths. Other methods, like RAP (Hao et al., 2023) and XoT (Ding et al., 2023) use planning approaches such as MCTS to refine reasoning trajectories.

**GFlowNets.** GFlowNets (Bengio et al., 2021) were developed to generate diverse, high-reward samples from unnormalized distributions (Shen et al., 2023b; Roy et al., 2023; Zhang et al., 2023c; Ma et al., 2024; Pan et al., 2023b), making them particularly effective in domains like molecule synthesis (Koziarski et al., 2024; Kim et al., 2024a; Lu et al., 2024) and biological sequence design (Ghari et al., 2023; Jain et al., 2022), where diversity is essential. Unlike traditional reinforcement learning (e.g., PPO), which focuses on maximizing reward, GFlowNets sample complete trajectories with probabilities proportional to their rewards, promoting exploration of the solution space. Recently, GFlowNets with LLMs have been applied to autoregressive tasks like token-level text generation (Hu et al., 2023a), but these approaches are limited to token-level sampling, making them less suited for complex reasoning. FOR extends GFlowNet principles to higher-level multi-step reasoning, modeling it as a Markovian flow through a DAG, enabling the exploration of diverse reasoning paths.

## 3 FOR FOR DIVERSE REASONING

### 3.1 MULTI-STEP LLM REASONING AS GENERATIVE FLOW

We start by formulating step-by-step LLM reasoning from the Markovian flow perspective. As we will show later, the new flow-based formulation allows us to connect LLM reasoning with the GFlowNet approaches for diversity-seeking finetuning. Meanwhile, the unique setting of multi-step LLM reasoning also inspires generalizations to the standard GFlowNets formalism (e.g., parameterization and exploration mechanisms) for enhanced efficiency. Figure 2 illustrates our approach. We refer to Appendix B for preliminaries and backgrounds of GFlowNets.

**The Multi-Step Reasoning Problem.** Consider a multi-step reasoning problem that gives an initial state $s_0$ and a goal $g$. For example, in BlocksWorld (Figure 2), an initial state is the starting configuration of the block stack, and a goal describes the desired configuration of blocks. Reasoning aims to find complete paths (or *trajectories*) that lead from the initial state to the states that satisfy

the goal. Given a current state $s$, applying an action on it leads to the transition to the next state $s'$, denoted as $s \to s'$. For example, in Figure 2, state $s_0$ transits to $s_1$ after an action `"pickup blue"`. A complete trajectory is thus a sequence of transitions $\tau = (s_0 \to s_1 \to \cdots \to s_n) \in \mathcal{T}$, where $s_n$ is the terminal state and $\mathcal{T}$ is the set of all complete trajectories. Given a current state $s_t$, there could be multiple alternative next actions, resulting in different branches of the reasoning. Also, different sequences of actions can lead to the same intermediate/terminal states, as shown in Figure 1. As a result, the multi-step reasoning has the structure of a directed acyclic graph (DAG).

The reasoning graph consists of diverse trajectories that lead to different terminal states. A crucial component often provided in reasoning tasks is the reward $R(s_n) \in \mathbb{R}_{\geqslant 0}$, which assigns a numerical value to any terminal state $s_n$. For instance, a terminal state meeting the goal $g$ receives a high reward. As discussed in §1, to generate diverse high-quality reasoning trajectories for solving a task, we want to sample the trajectories with probabilities *proportional* to the reward. This significantly differs from popular reinforcement learning methods (e.g., PPO) and prompting-based planning algorithms (e.g., RAP, ToT), which focus on optimizing for only the maximum-reward trajectory.

**The Flow Perspective.** Sampling complex multi-step trajectories from the (often unnormalized) reward is particularly challenging (LeCun et al., 2006; Qin et al., 2022). To overcome the difficulty, we consider the above reasoning problem from a flow-based viewpoint which was initially developed in (Bengio et al., 2021) and has been studied in other machine learning settings like molecule generation (Pan et al., 2022; Malkin et al., 2022a; Shen et al., 2023a; Li et al., 2023c; Lahlou et al., 2023; Li et al., 2024a; He et al., 2024). Specifically, we define a *trajectory flow* function $F : \mathcal{T} \to \mathbb{R}_{\geqslant 0}$. Analogous to the classical concept of flows in networks, the flow $F(\tau)$ can be thought of as the volume of water traveling along this path $\tau$. Based on this, for any state $s$, we can define the *state flow* $F(s) = \sum_{s \in \tau} F(\tau)$, and for any edge $s \to s'$, the *edge flow* $F(s \to s') = \sum_{s \to s' \in \tau} F(\tau)$. These concepts of (unnormalized) flow are connected to the (normalized) probability distributions. Specifically, the flow trajectory determines a distribution over trajectories:

$$P(\tau) = F(\tau)/Z, \quad Z = \sum_{\tau \in \mathcal{T}} F(\tau). \tag{1}$$

With a Markov assumption, it can be shown that the distribution factorizes into step-wise distributions:

$$P(\tau) = \prod_{t=1}^{n} P_F(s_t|s_{t-1}), \quad \text{where } P_F(s_t|s_{t-1}) = F(s_{t-1} \to s_t)/F(s_{t-1}). \tag{2}$$

That is, intuitively, $P_F(s_t|s_{t-1})$ characterizes the proportion of water at node $s_{t-1}$ that travels toward node $s_t$. The distribution $P_F$ is also called the *forward policy*, which can be used to generate a trajectory $\tau$ by sampling a sequence of transitions step-by-step starting from the initial state $s_0$. Equivalently (Bengio et al., 2023), there exists a *backward policy* that defines the distributions $P_B(\cdot|s_t)$ over the parents of each state $s_t$: $P_B(s_{t-1}|s_t) = F(s_{t-1} \to s_t)/F(s_t)$.

Let $\tau$ be the trajectory ending at the terminal state $s_n$. Recall that **our aim** in diverse LLM reasoning is to obtain a forward policy $P_F(s_t|s_{t-1})$ such that the resulting trajectory distribution is proportional to the reward. From the flow perspective, according to Eqs.(1) and (2), this aim is equivalent to approximating a Markovian flow $F$ such that $F(s_n)$ equals the reward (Bengio et al., 2021):

$$F(s_n) = R(s_n), \quad \forall \text{ terminal state } s_n. \tag{3}$$

The above flow-based concepts provide a rich set of constraints that can be converted into training objectives for learning the desired forward policy. For example, the *detailed balance* constraint $F(s_{t-1})P_F(s_t|s_{t-1}) = F(s_t)P_B(s_{t-1}|s_t)$ yields the respective objective used in molecule generation tasks (Bengio et al., 2023). In this work (§3.2), we devise the learning objective from the recent *trajectory balance* constraint shown to be more efficient (Malkin et al., 2022a). We consider the incorporation of other more recent extensions (Jang et al., 2023; Pan et al., 2023a) like subtrajectory balance (Madan et al., 2023) as future work.

**LLM Parameterization.** We parameterize the forward policy $P_F$ with an LLM and finetune as described in the next section. Specifically, for a reasoning task, we express its goal $g$, action $a$, and state $s$ as natural language (see Figure 1, BlocksWorld as an example). At each reasoning step $t$, the LLM generates an action $a_t \sim P_{\text{LLM}}(a|s_t; \theta, g, c)$, where $c$ is an appropriate prompt (e.g., instructions or in-context demonstrations). The prompts used in the experiments are detailed in Appendix C. Once an action is generated, the state transits to the next $s_{t+1} = T(s_t, a_t)$ with a transition function $T$.

Therefore, assuming that different actions applying to the same state $s_t$ lead to different next states, and that action $a_t$ leads to state $s_{t+1}$, we can write $P_F(s_{t+1}|s_t; \theta, g) = P_{\text{LLM}}(a_t|s_t; \theta, g, c)$. In the experiments, we follow previous work and define $T$ either by an LLM with appropriate prompts and greedy decoding (e.g., BlocksWorld as in (Hao et al., 2023)) or by the environment (e.g., Rubik's Cube as in (Ding et al., 2023)).

### 3.2 EFFICIENT DIVERSITY-SEEKING FINETUNING OF LLMs

The above new flow-based formulation of reasoning opens up the door for us to seamlessly import existing successful GFlowNet training methods for finetuning the LLM as the forward policy. These methods range from the *training objective* as mentioned earlier to the various *exploration strategies*, such as on-/off-policy sampling and local search (Kim et al., 2023; Zhang et al., 2022; Sendera et al., 2024), that substantially enhance the training efficiency. Algorithm 1 in Appendix D summarizes the FoR training procedure.

#### 3.2.1 TRAINING OBJECTIVE

In this work, we derive our training objective based on the trajectory balance approach (Malkin et al., 2022a), which has shown improved efficiency than other alternatives (Bengio et al., 2023; 2021). Specifically, for any complete forward trajectory $\tau = (s_0 \rightarrow s_1 \rightarrow \cdots \rightarrow s_n)$, the trajectory balance constraint, with a task goal $g$, says (Figure 2):

$$Z(s_0, g) \prod_{t=1}^{n} P_F(s_t|s_{t-1}; g) = F(s_n) \prod_{t=1}^{n} P_B(s_{t-1}|s_t; g), \qquad (4)$$

where we have used the fact that $P(s_n) = F(s_n)/Z(s_0, g)$ for the terminal state $s_n$. Plugging in the reward $R$, as motivated by Eq.(3), to provide supervision signals, the constraint leads to a loss function w.r.t the parameterized forward policy $P_F$:

$$l(\tau; \theta, g) = \left( \log \frac{Z(s_0, g) \prod_{t=1}^{n} P_F(s_t|s_{t-1}; \theta, g)}{R(s_n) \prod_{t=1}^{n} P_B(s_{t-1}|s_t; \theta, g)} \right)^2, \quad P_B(s_{t-1}|s_t; \theta, g) := \frac{1}{|\text{Pa}(s_t)|}, \qquad (5)$$

where $|\text{Pa}(s_t)|$ denotes the number of parents of state $s_t$, and Malkin et al. (2022a) suggested a canonical choice of setting $P_B(\cdot|s_t)$ to be uniform over the parents. Note that $Z$ is the total flow conditioning on each goal $g$ and initial state $s_0$. Estimating $\log Z$ can be cumbersome. We thus follow (Zhang et al., 2023a) to use the log-variance approximation, which implicitly estimates $\log Z$ given each trajectory $\tau$:

$$\Phi(\tau; \theta) = \log R(s_n) + \sum_{t=1}^{n} \log P_B(s_{t-1}|s_t; \theta, g) - \sum_{t=1}^{n} \log P_F(s_t|s_{t-1}; \theta, g), \qquad (6)$$

where $\Phi(\tau; \theta)$ equals to true $\log Z$ in the optimal case. Our optimization goal then turns into minimizing the variance of $\Phi(\tau; \theta)$ over different trajectories $\tau$ with the loss:

$$\mathcal{L}_V(\tau; \theta) = (\Phi(\tau; \theta) - \mathbb{E}_\tau[\Phi(\tau; \theta)])^2, \qquad (7)$$

where we draw sample trajectories $\tau$ from a behavior policy $\pi(\tau; \theta, g)$ for training, and $\mathbb{E}_\tau[\Phi(\tau; \theta)]$ is estimated with a mini-batch of sampled trajectories. Different configurations of $\pi$ result in on-policy, off-policy, and mixed explorations, which could impact training efficiency as shown in ablation studies (§4). We discuss our method of defining $\pi(\tau; \theta, g)$ below. If $\mathcal{L}_V(\tau; \theta)$ is globally optimized, the resulting flow satisfies Eq.(3) and $P_F(\cdot|\cdot; \theta, g)$ samples proportionally to the reward as desired.

#### 3.2.2 EFFICIENT EXPLORATION

The trajectory space is combinatorially large. We want to set up a $\pi(\tau, g; \theta)$ distribution in Eq.(7) that enables efficient exploration of the trajectory space and produces effective samples for training the parameters $\theta$ of the policy $P_F$. Drawing inspirations from the recent GFlowNet literature (Vemgal et al., 2023; Shen et al., 2023a; Hu et al., 2023a), we combine both on-policy and off-policy strategies. Moreover, we adapt the local search strategy from (Kim et al., 2023; Zhang et al., 2022; Sendera et al., 2024) to further enhance the exploration and yield stronger performance.

More specifically, for on-policy explorations, we use the online policy $P_F(s_t|s_{t-1}; \theta, g)$ itself and its tempered version to create training trajectories $\tau$ given the goal $g$ and initial state $s_0$ in a reasoning problem. For off-policy explorations, we use standard options from previous work (Vemgal et al.,

2023; Shen et al., 2023a; Hu et al., 2023a), including a replay buffer that prioritizes past high-reward trajectories, $\epsilon$-sampling, and offline trajectory data (for Game24 in §4.3). To further explore high-reward regions, we incorporate and modify a local search method (Figure 2) with higher efficiency. In particular, we select the trajectory with the highest reward in each trajectory batch, truncate the latter portion of the trajectory, and reconstruct it using a random policy $P_U$. This random policy avoids the computationally intensive forward process of LLMs, leading to enhanced efficiency. This approach reconstructs trajectories with a high probability of receiving higher rewards, as partially destroyed trajectories with high rewards are more likely to select the correct actions at the initial steps, while potentially making mistakes in the subsequent steps. Further details on the exploration strategies and local search process are provided in Appendices D and E, respectively.

## 4 EXPERIMENTS

### 4.1 EXPERIMENTAL SETTINGS

**Baselines.** We compare our approach with several prompting-based methods, including $k$-shot CoT (Wei et al., 2022) (with $k = 1, 5, 15$), ToT (Yao et al., 2024) (using BFS and DFS), GoT (Besta et al., 2024), XoT (Ding et al., 2023), and RAP (Hao et al., 2023). For fine-tuning-based methods, we evaluate SFT with diversity-enhancing decoding strategies like Temperature Sampling (Shi et al., 2024) ($\alpha = 1.0, 0.5, 0.1$, where $\alpha$ is a temperature used to adjust the probability distribution over the vocabulary for the next token), Nucleus Sampling (Holtzman et al., 2019) ($\eta = 0.95$, selecting from tokens that together make up $\eta$ of the probability mass), Typical Sampling (Meister et al., 2023) ($\tau = 0.95$, where the sampling distribution is restricted to words with negative log probabilities near the conditional entropy, and then $\eta$ of the distribution mass is truncated), and diverse beam search (Vijayakumar et al., 2016) (DBS, beam width $k = 20$). Additionally, we apply fine-tuning with PPO (Schulman et al., 2017) and GFN-CoT (Hu et al., 2023a). We also compare against OpenAI-O1, the latest and strongest reasoning model. All finetuning methods are trained on the same dataset as FOR. Except for Game24, which uses Llama-2-13B, Llama-3-8B is the base model for all other tasks. In the BlocksWorld task, we evaluated most baselines for a broad comparison, but some methods (e.g., GFN-CoT) showed suboptimal performance, which informed our decision to selectively apply baselines in the subsequent tasks. For the remaining tasks, we focused on high-performing methods from the initial evaluation.

**Evaluation.** As mentioned in §1, an effective reasoning method should not only produce correct solutions but also maximize the number of correct solutions found. Unlike previous approaches that evaluate a single solution per problem, we propose generating $n$ solutions per problem and assess methods based on four criteria: (1) **Accuracy (Acc)**: Success is defined if at least one of the $n$ solutions is correct. (2) **Diversity**: For solved problems, we report the average number of unique correct solutions among the $n$—higher is better (see Appendix C.1 for details). (3) **Creativity**: For each method, we report the proportion of unique successful trajectories found in all solutions that are not discovered by other methods. (See Appendix C.2 for details.) (4) **Runtime**: The average time taken by a method to produce one solution is used as an efficiency metric. For all the datasets, we recorded the average results and standard deviation of our performance from 3 repetitions, except for tree- and graph-structured methods and O1-series, which require significant time or expense to find a single solution. Creativity is calculated based on the result of 1 repetition due to the small standard deviation observed in accuracy and diversity metrics.

### 4.2 EMBODIED REASONING: BLOCKSWORLD

BlocksWorld involves a set of blocks with unique colors that can be stacked on top of each other or moved around. The goal of this task is to enable LLMs to plan a sequence of actions to transform an initial configuration of blocks into a desired goal configuration using a series of actions. The actions are text instructions (STACK, UNSTACK, PUT, PICKUP) generated based on domain rules and block positions, and a state is the current block orientation. Following (Hao et al., 2023), we use a second LLM (Llama-3-8b) aside from the policy model for state transition, which generates the next state $s_{t+1}$ given $(s_t, a_t)$ using greedy decoding.

**Setup.** Blocksworld examples (Valmeekam et al., 2024; Hao et al., 2023) are grouped by the minimum number of required actions: 30 examples for 2 steps, 57 for 4 steps, and 114 for 6 steps, following Hao et al. (2023). We select the first 15 of each group as the training examples for FOR and the rest

Table 1: Results on BlocksWorld, comparing *prompting-based* and *finetuning-based* methods on questions requiring two, four, and six steps, respectively. Standard deviations of three runs are shown in brackets (except for GPT-4o due to budget limit and ToT/RAP as they are exceedingly slow). We also report results from the *O1-series* models. Since these models are optimized for multi-step reasoning, their performance provides a reference for the upper limit of reasoning accuracy. (For O1-preview, we sampled only one solution due to budget limit.)

| Method | 2-step Acc. (%) | 4-step Acc. (%) | Diversity | Creativity (%) | 6-step Acc. (%) | Diversity | Creativity (%) | Runtime (s) (6-step) |
|---|---|---|---|---|---|---|---|---|
| *Prompting-based methods* | | | | | | | | |
| CoT (1-shot) | 48.88 (8.31) | 28.57 (5.83) | 1.05 (0.04) | 0.00 | 15.82 (2.08) | 1.05 (0.03) | 0.00 | 3.57 |
| CoT (5-shot) | 68.89 (8.31) | 42.86 (1.94) | 1.04 (0.03) | 0.00 | 29.63 (1.72) | 1.02 (0.01) | 0.00 | 3.68 |
| CoT (15-shot) | 64.44 (6.29) | 42.06 (4.89) | 1.03 (0.02) | 0.00 | 19.53 (1.26) | 1.03 (0.03) | 0.00 | 5.32 |
| CoT (GPT-4o, 1-shot) | 93.33 | 54.76 | 1.08 | 0.00 | 67.67 | 1.06 | 0.79 | 3.92 |
| ToT (BFS) | 13.33 | 14.28 | - | - | 5.05 | - | - | 398.74 |
| ToT (DFS) | 13.33 | 16.67 | - | - | 8.08 | - | - | 48.91 |
| RAP | **100.00** | 92.86 | - | - | 69.70 | - | - | 466.09 |
| *O1-series methods* | | | | | | | | |
| O1-mini (1-shot) * | 100.00 | 100.00 | 1.05 | 0.00 | 93.93 | 1.05 | 2.38 | 10.38 |
| O1-preview (1-shot) * | 100.00 | 95.24 | - | - | 78.79 | - | - | 36.61 |
| *Finetuning-based methods* | | | | | | | | |
| SFT ($\alpha$=1.0) | 44.44 (3.14) | 42.06 (5.44) | 1.05 (0.01) | 0.00 | 34.68 (2.52) | 1.04 (0.01) | 4.76 | 4.05 |
| SFT ($\alpha$=0.5) | 42.22 (3.14) | 39.68 (2.24) | 1.02 (0.02) | 0.00 | 29.63 (1.90) | 1.02 (0.02) | 0.79 | 4.07 |
| SFT ($\alpha$=0.1) | 26.67 (5.44) | 26.20 (3.89) | 1.00 (0.00) | 0.00 | 17.51 (1.26) | 1.00 (0.00) | 0.00 | 4.08 |
| SFT + DBS | 31.10 (3.11) | 38.88 (1.12) | 1.00 (0.00) | 0.00 | 18.85 (1.25) | 1.00 (0.00) | 0.00 | 15.71 |
| SFT + Nucleus | 48.89 (3.14) | 53.97 (2.97) | 1.04 (0.03) | 0.00 | 42.08 (1.71) | 1.12 (0.03) | 0.00 | 4.21 |
| SFT + Typical | 53.33 (5.44) | 48.41 (2.25) | 1.08 (0.02) | 0.00 | 37.71 (2.38) | 1.08 (0.02) | 0.00 | 3.65 |
| SFT + PPO | 46.66 (5.44) | 44.44 (2.24) | 1.11 (0.05) | 2.04 | 24.58 (1.72) | 1.08 (0.03) | 3.17 | 4.03 |
| SFT + GFN-CoT | 48.89 (8.81) | 44.42 (2.96) | 1.00 (0.00) | 0.00 | 40.73 (1.25) | 1.05 (0.03) | 0.00 | 4.08 |
| FoR (Ours) | **100.00** (0.00) | **98.41** (1.12) | **1.27** (0.02) | **12.24** | **78.44** (4.54) | **1.33** (0.03) | **9.52** | 13.98 |

as test examples. We sampled 8, 20, and 40 times for the 2, 4, and 6-step datasets, respectively, to report diversity and creativity. All the baselines are included in §4.1.

**Reward Design.** We compose a terminal state reward and an augmented intermediate reward to evaluate trajectories. Terminal state reward is assigned to a high positive value when a trajectory reaches the goal $g$. The augmented intermediate reward assesses actions by using the LLM to estimate the confidence of actions in achieving their goals. A natural choice is to use the log-likelihood of actions, $\log P_{\text{LLM}}(a_t|s_{t-1}, g)$. However, the value of $\log P$ is negative. To maintain monotonicity consistency and positive reward values, we use $-1/\log P_{\text{LLM}}(a_t|s_{t-1}, g)$ instead. The total reward is defined as: $R(s_n) = w \cdot \mathbb{I}(\text{success}) + \lambda \sum_{t=1}^{n} -1/\log P_{\text{LLM}}(a_t|s_{t-1}, g)$, where $w$ is the success weight (set to 100 and the following tasks).

**Results.** As shown in Table 1, our method demonstrates improvements across all metrics. In terms of accuracy, our method outperforms the best prompting-based baseline (GPT-4o with CoT) by 80% in 4-step tasks and 16% in 6-step tasks, and exceeds the best finetuning-based baseline (SFT + Nucleus) by 82% and 86%, respectively, highlighting its robustness with increasing task complexity. From a **diversity** standpoint, it outperforms SFT + PPO by around 14%, showing our method is able to generate more diverse solutions. Our method outperforms all other baselines in the **creativity** metric, discovering more unique solutions that are not found by other baselines. Most baselines do not find any unique solutions, resulting in a creativity score of 0. It is worth noting that the O1 series improves the accuracy to a large extent, but still struggles to find diverse reasoning paths. Additionally, FoR matches the inference speed of high-efficiency baselines like $k$-shot CoT, ToT (DFS), and SFT-based methods, while being 30× faster than ToT (BFS) and RAP on a single NVIDIA A100 GPU. Training costs are detailed in Appendix C.3.

### 4.3 MATH PUZZLE SOLVING: GAME OF 24

Game of 24 is a mathematical reasoning task that may have multiple solutions. The objective of this task is to use 4 integers and 4 basic arithmetic operations $(+, -, \times, \div)$ to reach 24, where each number can only be used once. Each action $a_t$ is defined as an equation composed of 2 numbers and an operator, and the state $s_t$ is the left number.

**Setup.** We use the LLM-reasoner dataset (Hao et al., 2024) and randomly select 20 examples for training, with examples ranked from easy to hard and those indexed 910-1010 used for testing. Since prior works show that LLMs struggle to online sample a correct trajectory in this task (Yao et al., 2024; Yu et al., 2023a), we use Python code to generate the offline ground-truth data with diverse trajectories, which is used for fine-tuning methods. In addition, to avoid the pitfalls of arithmetic

calculations with language models, we use Python code to calculate the results after "=" in an action $a_t$ across all methods evaluated in our experiment. We compare with baselines mentioned in §4.1 and additionally report the OpenAI-O1 mini performance. We sampled 20 times to report performance.

**Reward Design.** Similar to BlocksWorld, the success reward gives a high positive reward when a trajectory $\tau$ succeeds in reaching 24, and the augmented reward gives the product of the probability of correctness for each action $a_t$, given its last state $s_{t-1}$ provided by the untrained LLM model: $R(s_n) = w \cdot \mathbb{I}(\text{success}) + \prod_{t=1}^n P_{\text{untrained}}(a_t|s_{t-1})$.

**Results.** As shown in Table 2, FOR demonstrates superior accuracy and diversity in solving math puzzles compared to other baselines with the same base model. Not surprisingly, O1-mini and GPT-4o

Table 2: Results on Game of 24.

| Method | Acc. (%) | Diversity | Creativity (%) |
|---|---|---|---|
| *Prompting-based methods* | | | |
| CoT (5-shot) | 6.00 | 1.33 | 0.00 |
| CoT (GPT-4o, 5-shot) | **59.00** | **1.61** | **52.60** |
| XoT | 20.00 | - | - |
| ToT | 21.00 | - | - |
| RAP | 12.00 | - | - |
| *O1-series methods* | | | |
| OpenAI-O1-mini | 94.00 | - | - |
| *Finetuning-based methods* | | | |
| SFT ($\alpha = 1.0$) | 19.00 | 1.37 | 6.49 |
| FOR | 41.00 (0.82) | 1.52 (0.01) | 31.82 |

achieve better performance due to the stronger intrinsic mathematical knowledge and reasoning mechanism. We also investigate the fact that GPT-4o tends to use self-verification and reflection during Game24's inference. This may explain its superior performance in this task.

### 4.4 SPATIAL REASONING: RUBIK'S CUBE

The Rubik's Cube is a well-known puzzle requiring multi-step spatial reasoning. The model's task is to plan a sequence of rotations to restore a shuffled cube, where each state $s_t$ represents the block arrangement, and each action $a_t$ is a layer rotation (e.g., 90 or 180 degrees).

**Setup.** We randomly select 15 examples from the training dataset in (Ding et al., 2023), and evaluate different methods on a test set containing 183 examples. Each example can be solved in four steps. All the baselines are included in §4.1. For SFT, CoT and FOR, 10 solutions are sampled. See more details in Appendix C.6.

Table 3: Results on Rubik's Cube.

| Method | Acc. (%) | Diversity | Creativity (%) |
|---|---|---|---|
| *Prompting-based methods* | | | |
| CoT | 0.00 | 0.00 | 0.00 |
| CoT (GPT-4) | 1.09 | 1.00 | 4.35 |
| ToT (BFS) | 0.00 | - | - |
| GoT | 0.00 | - | - |
| XoT | 4.92 | - | - |
| *Finetuning-based methods* | | | |
| SFT ($\alpha = 1.0$) | 1.82 (0.06) | 1.00 (0.00) | 8.69 |
| SFT + PPO | 0.55 (0.45) | 1.00 (0.00) | 0.00 |
| FOR | **10.87** (1.18) | **1.29** (0.02) | **82.61** |

**Reward Design.** Similar to the above tasks, a high reward is given for successful restoration. The augmented reward is based on the difference in the minimum steps required to restore from the current cube state. If an action reduces the required steps, it receives a higher reward; otherwise, it gets a lower one. Formally, $R(s_n) = w \cdot \mathbb{I}(\text{success}) + \sum_{t=1}^n \exp(r(s_{t-1}) - r(s_t))$, where $r(s_t)$ represents the remaining minimum steps.

#### 4.4.1 RESULTS

As shown in Table 3, when there is a limited amount of training data available, our method outperforms all baselines in the Rubik's Cube task across all three metrics. It outperforms the best baseline (XoT) by over 120% in **accuracy**. Additionally, our approach generates 29% more **diverse** solutions, while other baselines only produce one solution on average. Notably, from a **creativity** perspective, our approach generates a large amount of unique solutions that other baselines are unable to discover.

### 4.5 ABSTRACTION REASONING: 1D-ARC

1D-ARC is a one-dimensional simplification of the ARC benchmark (Chollet, 2019), introduced in (Xu et al., 2023b). Each problem in the dataset contains a set of training input-output 1D grid pairs that capture a specific underlying rule and a test case that measures if the model correctly understands the rule. Following recent program search approaches (Wang et al., 2023b; Qiu et al., 2023; Butt et al., 2024), which frame the problem as a sequence of transformation functions, each action $a_t$ is a function (e.g., horizontal mirroring), with the intermediate grid as state $s_t$.

**Setup.** We randomly select 5 examples from the 1d_move_1p, 1d_padded_fill, and 1d_denoising tasks. These tasks involve moving the color bar by 1 pixel, filling in empty spaces enclosed by pixels,

and removing noise-like pixels, respectively. The 15 selected examples form the training set, while the remaining 45 examples from each task compose the test dataset. We sample 20 solutions for each example during inference. See Appendix C.7 for more details.

**Baselines.** Since there are no complex reasoning baselines (e.g., ToT) evaluated on this task, we compare against **Input-ouptut(IO) prompting** (Xu et al., 2023b; Mirchandani et al., 2023), which involves incorporating training input-output pairs into the prompt and prompting LLMs to infer output grids given the test input directly. **Program Only** and **Hypothesis Search** (Wang et al., 2023b) synthesize Python programs for transforma-

Table 4: Results on 1D-ARC.

| Method | Acc. (%) | Diversity | Creativity (%) |
|---|---|---|---|
| IO | 10.37 (1.21) | - | - |
| CoT | 39.51 (1.94) | 1.04 (0.01) | 1.45 |
| CoT (GPT-4o) | 40.00 | 1.00 | 0.00 |
| Program-Only | 0.74 | - | - |
| Hypo-Search | 1.48 | - | - |
| FOR | **50.37** (1.60) | **1.17** (0.02) | **21.74** |

tion, with the latter first generating language-based transformation hypotheses before program synthesis. Fine-tuning methods are not compared due to the lack of labeled reasoning data.

**Reward Design.** In addition to a success reward, we design the augmented rewards for actions based on how much they reduce the distance to the ground truth. Specifically, an action receives a higher reward if it reduces the hamming distance between the current state and the ground truth. The total reward is $R(s_n) = w \cdot \mathbb{I}(\text{success}) + \sum_{t=1}^{n} \sum_{i=1}^{K} \exp(h_d(s_{t-1}^i, g^i) - h_d(s_t^i, g^i))$, where $h_d(\cdot, \cdot)$ is hamming distance and $K$ is the number of training input-output pairs. $\mathbb{I}$ indicates 1 only when the searched program successfully transforms the test input to output.

### 4.5.1 RESULTS

As shown in Table 4, FOR substantially outperforms previous methods on all metrics, especially diversity and creativity. Previous approaches (hypothesis search and program-only) generate programs at once, which easily results in errors during the intermediate process, leading to inferior performance. As expected, FOR outperforms CoT, given CoT lacks mechanisms to try different solutions, causing them to rely solely on their internal knowledge and limiting their creativity.

### 4.6 LOGICAL REASONING: PRONTOQA

PrOntoQA is a logical reasoning task. Each test case includes a question (goal), a list of facts $\mathcal{A}$ (action space), and an initial state $s_0$. A state $s_t$ is the conclusion derived from the previous state $s_{t-1}$. Performance is evaluated using two metrics: prediction accuracy and proof accuracy. Prediction accuracy refers to the correctness of the final answer, regardless of the reasoning process. Proof accuracy, on the other hand, measures the correctness of the entire reasoning chain, ensuring that each step leading to

Table 5: PrOntoQA Results. *Pred Acc* measures the accuracy of the final conclusions, while *Proof Acc* evaluates the correctness of the reasoning process (e.g., no shortcuts/hallucinations).

| Method | In-Distribution | | Out-of-Distribution | |
|---|---|---|---|---|
| | Pred Acc.(%) | Proof Acc.(%) | Pred Acc.(%) | Proof Acc.(%) |
| *Prompting-based methods* | | | | |
| CoT | 52.20 (1.23) | 35.40 (1.86) | 43.50 (1.48) | 18.50 (1.91) |
| CoT (GPT-4o) | 89.00 | 47.80 | 62.92 | 24.78 |
| ToT (BFS) | 49.80 | 32.20 | - | - |
| RAP | 50.70 | 39.50 | - | - |
| *Finetuning-based methods* | | | | |
| STaR | 88.90 | 54.00 | 50.10 | 24.60 |
| FOR | 88.73 (1.33) | 54.60 (1.50) | **63.07** (1.71) | **28.88** (2.36) |
| FOR +STaR | **90.50** (1.89) | **54.70** (1.41) | 63.00 (2.13) | 26.67 (2.80) |

the final answer is accurate. Both metrics are calculated using rule-based string matching. The Diversity metric is not applicable, as each question has only one valid reasoning chain.

**Setup.** We randomly select 50 examples for the training set and 120 for the test set. The evaluation is conducted on both in- and out-of-distribution (OOD) examples, with 32 samples drawn per problem during inference. In addition to the baselines described in §4.1, we adopt STaR (Zelikman et al., 2022), which applies SFT on correct examples through online sampling. We also evaluate FOR on top of the model fine-tuned by STaR. See Appendix C.8 for more experimental details.

**Reward Design.** We removed the success reward to prevent the model from arriving at correct answers through flawed reasoning paths. Instead, we apply a rule-based augmented reward that evaluates the feasibility of a fact given the previous state $s_{t-1}$, checking if they share the same ontology. Formally, the reward is defined as $R(s_n) = \frac{1}{n} \sum_{t=1}^{n} w \cdot \mathbb{I}(s_{t-1}, s_t)$, where $w$ is a hyperparameter

Table 6: Ablation results on BlocksWorld for different components in FOR with Llama-3-8b.

| Method | 2-step | 4-step | | 6-step | |
| --- | --- | --- | --- | --- | --- |
| | Acc. (%) | Acc. (%) | Diversity | Acc. (%) | Diversity |
| FOR (Ours) | **100.00** (0.00) | **98.41** (1.12) | **1.27** (0.02) | **78.44** (4.54) | 1.33 (0.03) |
| - w/o local search | 100.00 (0.00) | 89.68 (2.97) | 1.18 (0.02) | 53.90 (2.10) | 1.31 (0.03) |
| - w/o augmented rewards | 100.00 (0.00) | 91.30 (1.10) | 1.22 (0.02) | 47.10 (1.30) | 1.21 (0.01) |
| - w/o replay buffer | 100.00 (0.00) | 94.44 (2.97) | 1.24 (0.04) | 72.38 (1.71) | 1.24 (0.01) |
| - w/o $\epsilon$-sampling | 100.00 (0.00) | 97.61 (1.95) | 1.26 (0.03) | 73.39 (2.38) | 1.25 (0.04) |

and $\mathbb{I}(s_{t-1}, s_t)$ is an indicator function. $\mathbb{I}(s_{t-1}, s_t)$ equals 1 only when the transition $(s_{t-1}, s_t)$ is part of the ground-truth reasoning path, ensuring no shortcuts are taken.

### 4.6.1 RESULTS

As shown in Table 5, FOR achieves superior results on both in- and out-of-distribution problems compared to all baselines. While FOR slightly outperforms the SFT-based STaR for in-distribution tasks, its advantage is far greater for out-of-distribution tasks. Moreover, combining FOR with STaR enhances in-distribution performance while preserving out-of-distribution success, revealing the complementary strengths of these methods.

### 4.7 ADDITIONAL ANALYSIS

**Ablation Study.** To further demonstrate the effectiveness of FOR, we conduct ablation studies to analyze the impact of individual components, focusing specifically on the BlocksWorld task. Table 6 summarizes the experimental results.

1) "Local search" significantly enhances the performance of FOR by improving exploration in the trajectory space and collecting high-reward trajectories for training. Removing local search results in a 31.3% decrease in 6-step task accuracy. 2) $\epsilon$-sampling also contributes to exploration, though to a lesser extent. 3) "Augmented intermediate rewards" play a critical role, as removing them leads to a 51% drop in 6-step accuracy and a 13% reduction in diversity. The left plot in Figure 3 shows the impact of varying the augmented reward weight $\lambda$. This improvement arises from distinguishing unsuccessful trajectories that are *more likely* and *less likely* to succeed, guiding the policy towards more successful paths by assigning higher probabilities to the former. However, when $\lambda$ becomes too large, accuracy declines as the success reward's influence diminishes. 4) the replay buffer contributes to FOR performance by leveraging historical high-reward trajectories for learning.

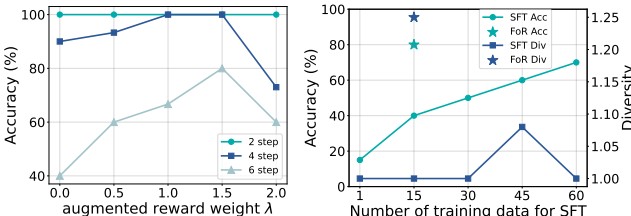

Figure 3: Additional Analysis on BlocksWorld. **Left**: Accuracy of FOR across different step settings with varying intermediate reward weight ($\lambda$). **Right**: Comparison of accuracy and diversity between SFT trained with varying data sizes and FOR trained with 15 examples on the test dataset.

**Data-Efficiency.** We tested the final 20 examples from the 6-step test set and adjusted the number of training examples for both FOR and the SFT methods. As illustrated in the right plot of Figure 3, SFT's accuracy improves with additional training data, while its diversity remains stable. However, SFT's performance remains lower than FOR's under any amount of training data. This is attributed to FOR's ability to learn from diverse reasoning trajectories, enhancing trajectory coverage and improving generalization to new cases.

## 5 CONCLUSION

We introduce FOR that efficiently trains LLM policy for diverse, high-quality reasoning paths with probability proportional to unnormalized reward. The core of the approach is the flow-based formulation of multi-step reasoning that allows us to adapt principled GFlowNet training strategies. On five representative tasks across embodied, math, logical, spatial, and abstraction reasoning, FOR show stronger performance and improved diversity than both finetuning-based and prompting-based baselines. We discuss limitations and broader impact in Appendix G and H.

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

# A ADDITIONAL RELATED WORK

**Reasoning with LLM.** Recent LLMs (Achiam et al., 2023; Touvron et al., 2023; Bai et al., 2022; Chowdhery et al., 2023) have demonstrated great potentials in tackling complex reasoning tasks (Cobbe et al., 2021; Mishra et al., 2022; Hendrycks et al., 2021; Rein et al., 2023; Mialon et al., 2023). **(1) Fine-tuning LLMs** is a primary way to enhance their reasoning abilities, including SFT and reinforcement learning (RL) approaches. **SFT** with large-scale and high-quality datasets of reasoning chains has proven very effective (Yu et al., 2023c; Yue et al., 2023; Yuan et al., 2024a). Various methods for constructing training samples have been proposed when ground truth reasoning chains are not available. For example, STaR (Zelikman et al., 2022) uses online sampling with self-correction to find positive samples. ReST$^{EM}$ (Singh et al., 2023) and V-STaR(Hosseini et al., 2024) filter samples with external verifiers. **RL** techniques, particularly reward-maximizing policy optimization methods like PPO, are widely employed in LLMs (Ouyang et al., 2022; Bai et al., 2022; Havrilla et al., 2024; Luong et al., 2024). However, both maximum likelihood training (i.e. SFT) and reward-maximizing policy optimization (e.g., PPO) do not encourage models to generate diverse solutions. **(2) prompting-based reasoning algorithms** aim to better elicit the knowledge inside LLMs without tuning their parameters. Techniques such as CoT (Wei et al., 2022) and its variants (Chen et al., 2022; Li et al., 2024b; Zhang et al., 2023e; Zhou et al., 2022; Kojima et al., 2022) have improved LLM performance by enabling them to generate intermediate steps before arriving at a final answer. To provide reasoning more guidance , self-evaluation (Xie et al., 2024b; Shinn et al., 2024; Madaan et al., 2024) and reward models are introduced to enhance reasoning process (Uesato et al., 2022; Lightman et al., 2023) Besides, a more relevant series of works combine LLM reasoning capabilities with planning and search algorithms such as MCTS (Hao et al., 2023; Feng et al., 2023; Zhao et al., 2024), tree and graph search (Jung et al., 2022; Zhu et al., 2022; Yao et al., 2024; Besta et al., 2024; Yao et al., 2023). Moreover, recent studies turn to amortizing computation for reasoning paths (Yuan et al., 2023; Wu et al., 2024; Bansal et al., 2024), such as self-correct (Kumar et al., 2024; Saunders et al., 2022), self-improvement (Tian et al., 2024; Yuan et al., 2024b).

**GFlowNets.** GFlowNets (Bengio et al., 2021) were originally proposed to learn policies for sampling from unnormalized distributions, with a primary motivation from scientific discovery (Jain et al., 2023a), which requires generating diverse high-reward samples (Shen et al., 2023b; Roy et al., 2023; Zhang et al., 2023c; Ma et al., 2024; Pan et al., 2023b), such as molecular generation (Koziarski et al., 2024; Kim et al., 2024a; Lu et al., 2024) and biological sequence generation (Ghari et al., 2023; Jain et al., 2022). Beyond the science domain, GFlowNets have also been applied in various downstream applications such as recommendation systems (Liu et al., 2023b), domain adaptation (Zhu et al., 2023), combinatorial optimization (Zhang et al., 2023b; Kim et al., 2024b) and explainability of deep neural networks (Li et al., 2023b). Additionally, GFlowNets have proven to be suitable for sampling from posterior distributions (Hu et al., 2023b; Deleu et al., 2022; 2024; Zhang et al., 2022). As a reinforcement learning method, prior works have incorporated intermediate feedback with GFlowNets to address sparse reward issues (Pan et al., 2023a; Jang et al., 2023; Pan et al., 2022) and multi-objective rewards (Jain et al., 2023b; Hernandez-Garcia et al., 2023; Chen & Mauch, 2023). There are also theoretical analyses treating GFlowNets as recurrent MCMC (Deleu & Bengio, 2023) and variational inference (Malkin et al., 2022b; Zimmermann et al., 2022) that are used to model the distribution over trajectories.

**Lateral and vertical thinking.** Vertical and lateral thinking (Waks, 1997; Ismayilzada et al., 2024) are two distinct approaches that differ significantly in their focus and methodology. Vertical thinking emphasizes logical, structured, and sequential reasoning, often following a step-by-step approach to solve problems. Our work aligns with this paradigm to generate multiple correct, structured reasoning processes to achieve specific goals. In contrast, lateral thinking prioritizes creativity and innovation, encouraging the exploration of unconventional perspectives and challenging established assumptions. Multiple benchmarks are proposed to evaluate the lateral thinking ability of LLMs (Huang et al., 2023; Chen et al., 2024a; Kraaijveld et al., 2024; Todd et al., 2024). To further improve the ability of lateral and divergent thinking, (Zhong et al., 2024) designs a fine-tuning and inference framework to generate unexpectable but reasonable answers, and (Summers-Stay et al., 2023) proposes a prompting framework to enhance such ability. In addition, riddle-solving QA tasks that require reasoning about unexpected or unconventional answers such as BrainTeaser(Jiang et al., 2023) and RiddleSense (Lin et al., 2021). Future work should investigate formalizing these tasks and developing quantitative approaches to effectively guide LLMs in tackling them.

# B  PRELIMINARIES AND BACKGROUND

GFlowNets (Bengio et al., 2021; 2023; Liu et al., 2023a) are a class of models that amortize the cost of sampling from an intractable target distribution over terminal states $\mathcal{X}$ by learning a neural network-facilitated approximation of the target distribution using its unnormalized density or reward function. The task of sampling from this distribution resorts to a decision-making process. Below we introduce GFlowNets with more details.

**Settings.**  We are given a pointed directed acyclic graph (DAG) $\mathcal{G} = (\mathcal{S}, \mathcal{A})$, where $S$ is a finite set of vertices (states), and $\mathcal{A} \subseteq \mathcal{S} \times \mathcal{S}$ is a set of directed edges (actions). If $s \rightarrow s'$ is an action, we say $s$ is a parent of $s'$ and $s'$ is a child of $s$. There is exactly one state that has no incoming edge, called the initial state $s_0 \in S$. States that have no outgoing edges are called *terminal states*. We denote by $\mathcal{X}$ the set of terminal states. A complete trajectory is a sequence $\tau = (s_0 \rightarrow \ldots \rightarrow s_n)$ such that each $s_i \rightarrow s_{i+1}$ is an action and $s_n = x \in \mathcal{X}$. We denote by $\mathcal{T}$ the set of complete trajectories and the terminal state as $\tau_x$.

Here we define the reward $R : \mathcal{X} \rightarrow \mathbb{R}^+$, and define a forward transition probability function, or a forward policy, $P_F(\cdot|s)$, which is a distribution over the children of every state $s \in S$. The forward policy is typically parametrized by a neural network that takes a representation of $s$ as input and produces the logits of a distribution over its children. Any forward policy $P_F$ induces a distribution over complete trajectories $\tau \in T$ (denoted by $P_F$ as well), which in turn defines a marginal distribution over terminal states $x \in \mathcal{X}$:

$$P_F(\tau) = P_F(s_0 \rightarrow \ldots \rightarrow s_n) = \prod_{i=0}^{n-1} P_F(s_{i+1}|s_i) \quad \forall \tau \in \mathcal{T} \tag{8}$$

Given a forward policy $P_F$, terminal states $x \in \mathcal{X}$ can be sampled from $P_F$ by sampling trajectories $\tau$ from $P_F(\tau)$ and taking their final states $s_n$. GFlowNets aim to find a forward policy $P_F$ such that the induced distribution $P_F^\top(x)$ is proportional to the reward function:

$$P_F^\top(x) \propto R(x) \tag{9}$$

**Training.**  Training GFlowNets considers achieving a consistent flow (Bengio et al., 2023; Malkin et al., 2022a), which means the flow for the forward direction should equal to the flow for the backward direction. Below we introduce relevant objectives.

**Detailed Balance (DB)**  The DB objective (Bengio et al., 2023) requires learning two objectives in addition to parametric forward policy $P_F(\cdot|s)$: 1. A *Backward policy*, which is distribution $P_B(s'|s; \theta)$ over the parents of any non-initial state. 2. A *State flow function*: $F(\cdot; \theta) : \mathcal{S} \rightarrow \mathbb{R}_{>0}$. Then DB loss for a single transition $s \rightarrow s'$ is defined as:

$$\mathcal{L}_{DB} = \left( \log \frac{F(s; \theta) P_F(s'|s; \theta)}{F(s'; \theta) P_B(s|s'; \theta)} \right)^2 \tag{10}$$

if $\mathcal{L}_{DB}$ is optimized to 0 for each transition, then the forward policy $P_F$ satisfies 9.

**Trajectory Balance (TB)**  Trajectory balance (Malkin et al., 2022a) introduces a backward policy $P_B$, which is a learned distribution $P_B(\cdot|s')$ over the parents of every state $s \in S$, and an estimated partition function $Z_\theta$ that is a scalar parameter describes the flow of initial state $Fs_0$ in the DB loss. The TB objective for a complete trajectory $\tau$ is defined as

$$\mathcal{L}_{TB}(\tau; \theta) = \left( \log \frac{Z_\theta \prod_{t=0}^{n-1} P_F(s_{t+1}|s_t; \theta)}{R(s_n) \prod_{t=0}^{n-1} P_B(s_t|s_{t+1}; \theta)} \right)^2 \tag{11}$$

If $\mathcal{L}_{TB}$ is made equal to 0 for every complete trajectory $\tau$, then 9 satisfies for all $x \in X$ and $Z$ is the inverse constant of proportionality: $Z = \sum_{x \in \mathcal{X}} R(x)$.

**Conditional GFlowNets.**  In a GFlowNet, both the policy and reward function can be conditioned on additional information. For instance, in the tasks we focus on, a GFlowNet policy generates actions sequentially for an embodied reasoning problem, starting from an initial state $s_0$ and a goal $g$. Furthermore, the allowable actions vary depending on the specific $s_0$ in each case. The conditional GFlowNets we train achieve amortization by sharing the policy model across different $s_0$ and $g$, enabling the model to generalize to initial states and targets that were not seen during training.

## C EXPERIMENTAL DETAILS

### C.1 DIVERSITY METRIC

We define the following metric to measure the diversity of reasoning paths found by different approaches. Under the same number of samplings at inference time, we count the number of different successful trajectories a policy finds for the successful example on average.

$$\text{Diversity} = \frac{\sum_{i=1}^{n} S_i \cdot \mathbb{I}(S_i \geqslant 1)_i}{\sum_{i=1}^{n} \mathbb{I}(S_i \geqslant 1)_i} \geqslant 1 \tag{12}$$

where $n$ is the total number of problems, $S_i$ is the number of successful trajectories found for the $i$-th question, and $\mathbb{I}(S_i \geqslant 1)$ is an indicator function that is 1 if there is at least one successful trajectory found for the $i$-th question and 0 otherwise. Thus, the denominator is the number of examples in which a model finds at least one trajectory, and the nominator is the sum of all successful trajectories a model finds across all examples. The smallest diversity is 1 when a method can only find at most one successful trajectory on average, and diversity $= 1.5$ indicates a method is able to find 1.5 different successful trajectories on average.

### C.2 CREATIVITY METRIC

We define the following metric to quantify the creativity of a reasoning method. Given the same number of samples during inference, we calculate the ratio of unique successful trajectories that a method identifies in the test dataset $D_{test}$, which are not found by any other methods. Let $\mathcal{M} = \{m_1, m_2, \ldots, m_{|\mathcal{M}|}\}$ represent the set of reasoning methods. For the $i$-th problem, the $l$-th method has a solution set $S_i^l$, where $1 \leqslant l \leqslant |\mathcal{M}|$. The complete set of solutions across all methods is defined as:

$$S = \bigcup_{i=1}^{n} \bigcup_{l=1}^{|\mathcal{M}|} S_i^l \tag{13}$$

Then we can define the creativity metric of method $m_l$ as:

$$\text{Creativity}(m_l) = \frac{1}{|S|} \sum_{i=1}^{|D_{\text{test}}|} \sum_{s \in S_i^l} \mathbb{I}(s, i, l), \tag{14}$$

where for the $i$-th problem, if the solution $s \in S_i^l$ is found only by method $m_l$ and not by any other method $m_k$ (where $k \neq l$), then $\mathbb{I}(s, i, l) = 1$. Otherwise, $\mathbb{I}(s, i, l) = 0$. The indicator function $\mathbb{I}(s, i, l)$ is defined as:

$$\mathbb{I}(s, i, l) = \begin{cases} 1, & \text{if } s \notin \bigcup_{k \neq l} S_i^k \\ 0, & \text{otherwise} \end{cases} \tag{15}$$

### C.3 EFFICIENCY ANALYSIS

All experiments were conducted using a server with a single NVIDIA A100 GPU. Below we report the average of 3 times training for 6-step training cost on BlocksWorld dataset for 10 epochs. We compare with SFT, PPO and table 7 shows the results.

PPO and FOR need much more training costs because they need exploration and interaction with environments to collect trajectories for training, and SFT only trains on ground-truth trajectories which take less time.

Table 7: Training time shown is seconds when training on the BlocksWorld.

| Method | Runtime (s) |
|---|---|
| SFT | 196.37 |
| SFT+PPO | 1740.96 |
| FOR | 6833.37 |

### C.4 BLOCKSWORLD.

**FoR Setup.** During the training, we finetune the LLM with LoRA (Hu et al., 2021) with $r = 32$, $\alpha = 64$, and dropout=0.1. We set $\epsilon$ from 0.3 and decrease it to 0.01, $\beta$ from 1 to 2, and the probability $\delta$ using replay buffer increases from 0.3 to 0.5 throughout the iterations linearly. The learning rate is set to

Table 8: OOD results on BlocksWorld.

| Method | 2-step to 4-step | | | 4-step to 6-step | | |
|---|---|---|---|---|---|---|
| | Acc. (%) | Diversity | Creativity (%) | Acc. (%) | Diversity | Creativity (%) |
| CoT (1-shot) | 9.52 | 1.0 | 3.12 | 2.02 | 1.0 | 0 |
| CoT (5-shot) | 14.28 | 1.0 | 3.12 | 12.12 | 1.08 | 3.45 |
| CoT (15-shot) | 11.90 | 1.0 | 3.12 | 8.08 | 1.0 | 0 |
| ToT (BFS) | 9.52 | - | - | 8.08 | - | - |
| ToT (DFS) | 4.76 | - | - | 6.06 | - | - |
| RAP | **80.95** | - | - | 34.34 | - | - |
| SFT ($\alpha$=1.0) | 11.92 | 1.0 | 9.37 | 28.28 | 1.03 | 1.15 |
| FOR (Ours) | 71.43 | **1.20** | **59.38** | **65.65** | **1.25** | **60.92** |

Table 9: Baseline results with diversity-encouraging instruction prompt.

| Method | 4-step | | | 6-step | | |
|---|---|---|---|---|---|---|
| | Acc. (%) | Diversity | Creativity (%) | Acc. (%) | Diversity | Creativity (%) |
| CoT (1-shot) | 16.67 (-10.90) | 1.00 (-0.05) | 0.0 (0.00) | 11.11 (-4.71) | 1.09 (+0.04) | 0.0 (0.00) |
| CoT (5-shot) | 59.52 (+16.66) | 1.12 (+0.08) | 2.04 (+2.04) | 33.33 (+3.70) | 1.03 (+0.00) | 0.79 (+0.79) |
| CoT (15-shot) | 52.38 (+12.32) | 1.09 (+0.06) | 0.0 (0.00) | 13.13 (-6.40) | 1.07 (+0.04) | 0.0 (0.00) |
| SFT ($\alpha$=1.0) | 59.52 (+17.46) | 1.10 (+0.05) | 0.0 (0.00) | 47.47 (+12.79) | 1.10 (+0.06) | 0.0 (0.00) |
| FOR (Ours) | **98.41** | **1.27** | **12.24** | **78.44** | **1.33** | **9.52** |

1e-4 with a cosine annealing schedule, and the number of training iterations is set to 10. Reward weight $\lambda$ is set to 1.5. In our ablation study when setting $\lambda = 0$, we add a small number $b = 0.5$ to avoid $\log 0$. Table 4 shows the template we use for the forward policy in the 6-step setting, and its difference between 2-step and 4-step is only replacing the 6-step demonstration to 2-step and 4-step. During testing, we sample 8, 20, and 40 trajectories for 2, 4, and 6 steps respectively. As long as one trajectory reaches the goal, we label this instance as solved, all the baselines conform to the same rule.

**Additional details for baselines.** We compare FOR the following baselines:

(1) *Chain-of-Thoughts prompting (CoT)* (Wei et al., 2022): It concatenates $k$ problems with ground truth solutions and the test problem, and prompts the LLM to generate a solution. We test the setting where $k = 1, 5, 15$, and pass the test cases to LLMs at the same times as FOR, and the test case is regarded as solved if at least one plan is correct.

(2) *Tree-of-Thoughts prompting (ToT)* (Yao et al., 2024): This approach constructs a tree of actions and searches for the solution with the highest reward. For each action, the reward includes (a) the likelihood of the LLM predicting the action and (b) self-evaluation, where the LLM is prompted with the question, "Is this action good?" and the answer is mapped to a reward value. We implement ToT with both breadth-first search (BFS) and depth-first search (DFS), terminating after generating 10 solutions.

(3) *Reasoning-via-Planning (RAP)* (Hao et al., 2023): This method also conducts a tree search for the optimal solution. Different from ToT, it alternatively predicts the next action and predicts the resulting block arrangement. Besides the rewards used in ToT, if the predicted block arrangement matches the goal, a high reward will be assigned.

(4) *Supervised Fine-Tuning (SFT)*: We use problems in the training set and their corresponding ground truth solutions to finetune the LLM. Note that this is an easier setting than FOR which does not have access to ground truth solutions. We train LLM with the same iterations as FOR.

(5) *Proximal Policy Optimization (PPO)* (Schulman et al., 2017): This is a widely-used reinforcement learning method for LLM training. We design the objective to encourage the LLM to generate solutions that satisfy the goal. Following the common practice of previous work (Ouyang et al., 2022; Wang et al., 2023a), we penalize the policy if it deviates too much from the reference policy. Formally, the objective is $\max_{\pi_\theta} \mathbb{E}_{\tau \sim \pi_\theta} [R(x, y)] - \beta \mathbb{D}_{\text{KL}} [\pi_\theta(y \mid x) \| \pi_{\text{ref}} (y \mid x)]$.

(6) *GFN-CoT* (Hu et al., 2023a): This approach adapts the GFlowNets training paradigm, which is a diversity-seeking RL method, to enable posterior sampling of the intermediate reasoning process from LLMs.

---

**Prompt for BlocksWorld**

I am playing with a set of blocks where I need to arrange the blocks into stacks.
Here are the actions I can do

Pick up a block
Unstack a block from on top of another block
Put down a block
Stack a block on top of another block

I have the following restrictions on my actions:
I can only pick up or unstack one block at a time.
I can only pick up or unstack a block if my hand is empty.
I can only pick up a block if the block is on the table and the block is clear.
A block is clear if the block has no other blocks on top of it and if the block is not picked up.
I can only unstack a block from on top of another block if the block
I am unstacking was really on top of the other block.
I can only unstack a block from on top of another block if the block I am unstacking is clear.
Once I pick up or unstack a block, I am holding the block.
I can only put down a block that I am holding.
I can only stack a block on top of another block if I am holding the block being stacked.
I can only stack a block on top of another block if the block onto which I am stacking the block is clear.
Once I put down or stack a block, my hand becomes empty.

- - - - - - - - - - - - - - - - - - - - - - - - - - - - - - - - - - - - - - - - - - - - -

[STATEMENT]
As initial conditions I have that, the orange block is clear, the hand is empty, the red block is on top of the
blue block, the orange block is on top of the red block and the blue block is on the table.
My goal is to have that the blue block is on top of the orange block.
My plan is as follows:
[PLAN]
unstack the orange block from on top of the red block
put down the orange block
unstack the red block from on top of the blue block
put down the red block
pick up the blue block
stack the blue block on top of the orange block
[PLAN END]

- - - - - - - - - - - - - - - - - - - - - - - - - - - - - - - - - - - - - - - - - - - - -

[STATEMENT]
As initial conditions I have that, <current state>
My goal is to My goal is to have that <goals>
My plan is as follows:
[PLAN]
<action>

Figure 4: Prompt template for the embodied reasoning task (6-step).

**Performance on OOD settings.** We further assess performance on out-of-distribution (OOD) settings. Specifically, we train the model using FoR and SFT on a 2-step training set and evaluate them on a 4-step test set, and train the model on the 4-step training set and evaluate them on the 6-step test set. This allows us to analyze their generalization on OOD problems. For prompting-based baselines, we use 2-step and 4-step examples as demonstrations, respectively.

According to the result in table 8, FoR maintains the highest accuracy (71.43%) on OOD tasks compared to other methods like CoT and SFT, which range from 9.52% to 14.28%. FoR also achieves greater diversity (by an absolute improvement of 0.2 over SFT), highlighting its superior generalization and solution exploration capabilities.

**Additional baseline results with diversity-encouraging instruction.** To further stimulate the diverse problem-solving ability in the baseline approaches, we add a diversity-encouraging prompt as instruction:

Table 10: An example of the probability of two trajectories to be sampled by FOR.

| Initial State | goal | Trajectory | Terminal State | $P_F(\tau)$ |
|---|---|---|---|---|
| | the red block is on top of the blue block | 1. unstack the orange block from on top of the blue block 2. put down the orange block 3. unstack the blue block from on top of the red block 4. put down the blue block 5. pick up the red block 6. stack the red block on top of the blue block | | 0.47 |
| | the red block is on top of the blue block | 1. unstack the orange block from on top of the blue block 2. put down the orange block 3. unstack the blue block from on top of the red block 4. stack the blue block on top of the orange block 5. pick up the red block 6. stack the red block on top of the blue block | | 0.46 |

*Please carefully understand the goals and initial states, then come up with diverse solutions and think outside the box.*

We evaluate multiple baseline methods using LLama-3-8B as the base model, following the exact same settings as in Section 4.2. The results for BlocksWorld are reported in Table 9. The numbers in parentheses indicate the performance difference compared to the original prompt without the diversity-encouraging instruction.

We observe that diversity-encouraging prompts for the CoT and SFT baselines lead to improvements in both diversity and accuracy, with average absolute gains of 0.03 and 5.11%, respectively. However, FoR still outperforms them, achieving average absolute improvements of 0.19 in diversity, 9.46% in creativity, and 34.93% in accuracy compared to the best baseline for each metric.

**Additional case study.** In Table 10, we show examples generated by FOR. We observe that after training, FOR can sample the terminal state with probability approximately proportional to the rewards, leading to an approximate sampling of different plans with the same probability. This empirically verifies the efficacy of the training objective.

### C.5 GAME OF 24.

**FOR Setup.** See Figure 5 for the prompt template used in the experiment of the Game of 24.

We use LoRA to train the model with $r = 8$, $\alpha = 32$, dropout=0.1. We load the LLM in fp16, and set the hyperparameters as follows: batch size = 4, learning rate = 1e-5, number of epochs = 5, and the reward weight $w = 100$.

### C.6 RUBIK'S CUBE

**FOR Setups.** The training hyperparameters are identical to BlocksWorld. During testing, we sample 10 trajectories. See Figure 6 for the prompt template of the Rubik's Cube task.

**Additional details for baselines.** Apart from the baselines in Blocksworld, we further compare them with GoT and XoT.

*Graph-of-Thought(GoT)* (Besta et al., 2024): GoT builds upon the ToT method by introducing the ability to create graph-like thought structures, achieved through the aggregation and refinement of thoughts during intermediate search stages. While this approach allows for more adaptable

---

Prompt for Game of 24

Use numbers and basic arithmetic operations (+ - * /) to obtain 24.
For each step, you are only allowed to choose two of the remaining numbers to obtain a new
number.
Input: 4 4 6 8
Steps:
4 + 8 = 12 (left: 4 6 12)
6 - 4 = 2 (left: 2 12)
2 * 12 = 24 (left: 24)
Input: 2 9 10 12
Steps:
12 * 2 = 24 (left: 9 10 24)
10 - 9 = 1 (left: 1 24)
24 * 1 = 24 (left: 24)
Input: 4 9 10 13
Steps:
13 - 10 = 3 (left: 3 4 9)
9 - 3 = 6 (left: 4 6)
4 * 6 = 24 (left: 24)
Input: 1 4 8 8
Steps:
8 / 4 = 2 (left: 1 2 8)
1 + 2 = 3 (left: 3 8)
3 * 8 = 24 (left: 24)
Input: 5 5 5 9
Steps:
5 + 5 = 10 (left: 5 9 10)
10 + 5 = 15 (left: 9 15)
15 + 9 = 24 (left: 24)
Input: <input>
Steps:
<action>

Figure 5: Prompt template for the mathematical puzzle task.

---

Prompt for Rubik's Cube

You are a virtual expert in solving a 2x2 Pocket Cube. Your task is to restore a scrambled 2x2 Rubik's
Cube to its original state. All the given problems can be solved in 1 to 4 moves. You cannot exceed more
than 11 moves. Provide the sequence of moves required for the restoration. Please follow the instructions
and rules below to complete the solving:
1. A 2x2 Pocket Cube has six faces, namely: [Upper, Front, Bottom, Left, Right, Back] Each consisting of
a 2x2 grid of squares, with each square having its own color.
2. Colors in the Cube are represented in numbers: [0, 1, 2, 3, 4, 5]
3. You must make a move to the Cube to achieve a Restored State. Note that we just need each face to
have the same numbers, no matter which face has which color.
4. A restoration of a Pocket Cube is to move squares in each face to have the same numbers.
5. You are only allowed to use the following moves [U, U', U2, R, R', R2, F, F', F2].
Now strictly follow the above process to form Restoration Moves.

- - - - - - - - - - - - - - - - - - - - - - - - - - - - - - - - - - - - - - - - - - - - - - - - -

[STATEMENT]
As initial state of the cube, I have that
[Initial Cube State]:
<current state>
[Process]:
[Step 1]
[Move] <action>

Figure 6: Prompt template for the spatial Reasoning task.

thought structures, it still requires several LLM inference calls for evaluation, leading to substantial
computational expenses.

---

**Prompt for 1D-ARC**

You are provided with a series of input-output pairs, where each value from 'a' to 'j' represents a different color, and '.' denotes a blank cell. For example, [['.','a','.'],['.','.','b']] represents a grid with 2 rows and 3 columns, where color 'a' is at position (1,0) and color 'b' is at position (2,1).
Coordinates are expressed in 2D positions (row, col), with 'row' indicating the row number and 'col' indicating the column number, both using zero-based indexing. The input-output pairs may not cover all possibilities, so you should infer the simplest possible relationship between them.
Your task is to reason through a sequence of Python functions that can transform the input grid into the output grid. Please strictly follow this process to form the appropriate Python function.
[STATEMENT]
You have the following input-output pairs:
[Initial Grid State]:
<init_state>
Based on the provided list of Python functions, select the appropriate function to achieve the transformation from the input to the output:
<python_function>
Now, please choose one function from the above list:
<action>

---

Figure 7: Prompt template for abstraction reasoning task.

*Everything-of-Thought(XoT)* (Ding et al., 2023): XOT is a collaborative framework combining LLMs with MCTS to optimize the thought generation process, aiding LLMs in solving complex problems. It first trains a small network to explore the space fast while LLMs refine and correct the thoughts generated by MCTS.

### C.7   1D-ARC

**FOR Setups.** Except that we train the model for 1 iteration, other training hyperparameters are identical to BlocksWorld. We use the hand-crafted transformation functions in ARC Challenge 2nd-place (de Miquel, 2021) on Kaggle 2020. See Figure 7 for the prompt template of the 1D-ARC task. Part of the prompt is adapted from (Tan & Motani, 2023). For CoT and FOR, we sampled 20 times. IO methods directly predict the output grids without an explicit reasoning process, while program-only and Hypothesis Search approaches generate a large number of candidate programs and choose the best candidates, which is time-consuming. As a result, we do not report diversity and creativity metrics for these methods.

**Additional details for baselines.** In addition to IO and CoT, we also compare our approach with Hypothesis Search which belongs to discrete program search methods (Barke et al., 2024; Xu et al., 2023a; Lee et al., 2024).

*Hypothesis Search* (Wang et al., 2023b): The method first generates multiple hypotheses describing the underlying transformation rules in natural language, and then selects a subset of potentially correct hypotheses. Based on these selected hypotheses, numerous Python programs are synthesized, which are subsequently tested on the training input-output pairs to verify whether they pass all the cases. If a program successfully passes all the training input-output pairs, it is considered to have accurately captured the underlying transformation rules.

### C.8   LOGICAL REASONING.

**OOD data creation**  We separate the in-distribution and OOD data by topics and ontology. We use the animal-related problems as in-distribution examples and the number-related problems as OOD examples.

**Setup.**  We use LoRA to train the model with $r = 8$, $\alpha = 32$, dropout=0.1. We load the LLM in fp16, and set the hyperparameters as follows: batch size = 4, learning rate = 5e-6, number of epochs = 40, and the reward weight $w = 100$. See Table 8 for the prompt template of the logical reasoning task.

---

**Prompt for PrOntoQA**

Given a list of facts, and a current claim, output one possible fact as the next step ONLY BASED ON THE LAST CLAIM without using your knowledge. Be sure to copy the EXACT sentence in the facts. Do NOT change any wording. Do NOT create your own words. Give me the next step ONLY.

- - - - - - - - - - - - - - - - - - - - - - - - - - - - - - - - - - - - - - - - - - - - - - - - -

Facts 1: Each lepidopteran is an insect. Each arthropod is a protostome. Every animal is multicellular. Protostomes are invertebrates. Each whale is bony. Each painted lady
is a butterfly. Invertebrates are animals. Butterflies are lepidopterans. Each insect is six-legged. Every insect is an arthropod.
Arthropods are not bony.
Query 1: True or false: Sally is not bony.
Claim 1.1: Sally is an insect.
Next 1.1: Every insect is an arthropod.
Claim 1.2: Sally is an arthropod.
Next 1.2: Arthropods are not bony.
Claim 1.3: Sally is not bony.
Next 1.3: Finish.

- - - - - - - - - - - - - - - - - - - - - - - - - - - - - - - - - - - - - - - - - - - - - - - - -

Facts 2: Lepidopterans are insects. Every animal is multicellular. Each insect is an arthropod. Each invertebrate is an animal. Insects are six-legged. Arthropods are small. Arthropods are invertebrates. Each butterfly is a lepidopteran. Whales are not small.
Query 2: True or false: Polly is not small.
Claim 2.1: Polly is a lepidopteran.
Next 2.1: Lepidopterans are insects.
Claim 2.2: Polly is an insect.
Next 2.2: Each insect is an arthropod.
Claim 2.3: Polly is an arthropod.
Next 2.3: Arthropods are small.
Claim 2.4: Polly is small.
Next 2.4: Finish.

- - - - - - - - - - - - - - - - - - - - - - - - - - - - - - - - - - - - - - - - - - - - - - - - -

Facts 3: <facts>
Query 3: <query>
Claim 3.1: <initial state>
Next 3.1: <action>

---

Figure 8: Prompt template for logical reasoning task.

**Additional details for Baselines.** Apart from CoT, ToT, and RAP, we compare FOR with STaR (Zelikman et al., 2022), which uses online sampling to filter our positive examples consistent with ground truth trajectories to finetune the LLM. Note that this is an easier setting than FOR which doesn't have access to ground truth solutions. It also indicates an upper bound of SFT methods that do not rely on ground truth solutions, like. All baselines use Llama3 8B as the base model.

## C.9 GSM8K.

In addition to the above datasets, we additionally evaluate the proposed FOR on GSM8K (Cobbe et al., 2021). We follow RAP (Hao et al., 2023) to define an action as an intermediate sub-question to solve the problem and a state as all the history intermediate pairs of a sub-question and its answer. We conduct experiments with 2-shot settings and compare them with supervised fine-tuning (SFT), CoT, CoT with self-consistency (CoT-SC), and RAP. For each problem, we sample 4 solutions and success is indicated as long as 1 solution is correct. For training, we construct the training dataset with the last 50 examples in the GSM8K training set. The implementation of baselines refers to (Hao et al., 2024). Due to the lack of established evaluation metrics for assessing the diversity of open-ended mathematical reasoning, we manually annotate 50 test examples to evaluate the similarity between reasoning trajectories, determining whether two reasoning trajectories are semantically equivalent or not.

Table 11: Results on GSM8K.

| Method | Acc. (%) | Diversity |
|---|---|---|
| CoT | 45.72 | 1.12 |
| CoT-SC | 41.74 | - |
| RAP | 37.16 | - |
| SFT ($\alpha = 1.0$) | 52.69 | 1.13 |
| **FoR** | **57.39** | **1.26** |

The results are shown below: FoR shows effectiveness on GSM8K and exceeds the accuracy and diversity of all baselines, which demonstrates the potential of FoR for extending to more open-ended reasoning tasks.

As shown in the table 11, in the task of GSM8K, FoR outperforms all baselines in both accuracy (by an absolute improvement of 4.7% over SFT) and diversity (by an absolute improvement of 0.13 over SFT). These results highlight the potential of FoR for extending to more open-ended reasoning tasks.

## D EXPLORATION AND TRAINING

FoR employs the following techniques to explore during the training phase:

1. Online training: (1) we employ the online policy $P_F(a_t|s_{t-1}, \alpha)$, and its tempered version (2) Similar to $\epsilon$-greedy, we sample action at step $t$ by $P_F$ with probability $\epsilon$, and sample with uniform distribution over action space $P_U(a_t|s_{t-1})$ with $(1 - \epsilon)$ probability. (3) To further explore the high-reward region, we modified the local search (Kim et al., 2023; Zhang et al., 2022). More specifically, we select the trajectory with the highest reward in a batch and conduct a destroy and reconstruction process for augmenting the trajectories to enable a higher probability of sampling successful trajectories, referring to Appendix E for more details.

2. Offline training: (1) Experience replay represents a significant advancement in reinforcement learning, offering enhancements in both learning efficiency and stability, as evidenced by recent empirical studies in GFlowNets (Vemgal et al., 2023; Shen et al., 2023a). To optimize the utility of the trajectories collected, we set up a prioritized replay buffer (PRB). This buffer facilitates the sampling of trajectories in proportion to their reward value, $R(\tau)$, or its logarithmic value, thereby prioritizing potentially more informative experiences. (2) For tasks (e.g. Game of 24) that have a large space, online sampling diverse trajectories with LLMs is computationally expensive. Therefore, we integrate the offline trajectories to have a larger coverage of space and improve the efficiency, which means $\delta = 0$.

Algorithm 1 describes the training framework.

---

**Algorithm 1** FoR Training

---

1: **Input:** $I$: number of iterations, $P_F$: initial LLM policy, $\mathcal{D}$: Prioritized Replay Buffer. $M$: Batch-size, $\delta$: online-offline ratio, $\mathcal{E}$: Training Dataset, $\mathcal{O}$: offline Data
2: **Output:** Trained policy $P_F$.
3: **for** $i = 1$ to $I$ **do**
4:     Sample from training dataset $\mathcal{E}$ with initial state $s_0$ and goal $g$
5:     Sample from $u \sim [0, 1]$
6:     **if** $u < \delta$ **then**
7:         // **Exploration**
8:         Sample $M$ online trajectories $\{\tau_1, ..., \tau_M\}$ with forward policy $P_F$
9:         Select trajectory $\tau_m \in \{\tau_1, ..., \tau_M\}$ with the largest $R(\tau_m)$
10:        $\{\tau'_{1'}...\tau'_{N'}\} \leftarrow$ Local Search$(\tau_m)$
11:        Update $\mathcal{D} \leftarrow \mathcal{D} \cup \{\tau_1, ..., \tau_M\} \cup \{\tau'_{1'}...\tau'_{N'}\}$
12:     **else**
13:         // **Exploitation**
14:         **if** is Game24 **then**
15:             sample $M$ offline trajectories from Offline Data $\mathcal{O}$
16:         **else**
17:             sample $M$ offline trajectories from $\mathcal{D}$
18:         **end if**
19:     **end if**
20:     Exploit $M$ (with $N'$) trajectories to compute objective function in Eq 7.
21:     Update the parameter in $P_F$ with respect to Eq 7
22: **end for**
23: **return** $P_F$

---

# E    MODIFIED LOCAL SEARCH

Local search is a simple data augmentation technique for GFlowNets (Kim et al., 2023; Zhang et al., 2022; Sendera et al., 2024), which is designed to enhance training efficiency. Different from the original local search which is conducted on each sampled trajectory, we select the trajectory in a batch with the highest reward to conduct a local search. Here we denote the trajectory reward $R(\tau)$ as the reward of the terminal state of the trajectory $R(\tau = (s_0 \to \ldots \to s_n)) = R(s_n)$. More specifically, we illustrate our modified local search for one instance as follows:

- **Sampling:** Sample a set of complete trajectories $\{\tau_1, ..., \tau_M\}$ using forward policy $P_F$ and select the $\tau_m$ with the largest reward $R(\tau_m)$

- **Searching:** We destroy $\tau_m$ by backtracking $K$-step into a partial trajectory and reconstruct the complete trajectory from the partial trajectory:

$$\tau_{destroy} = (s_0 \to \ldots \to s'_{n-K}), \qquad \tau_{recon} = (s'_{n-K} \to \ldots \to s'_n) \tag{16}$$

We obtain the local searched trajectory $\tau'$:

$$\tau' = (s_0 \to \ldots \to s'_{n-K} \to \ldots \to s'_n) \tag{17}$$

Where the $\tau_{recon}$ is completed by the random policy $P_U$ which randomly selects a feasible action for efficiency. We can obtain a set of reconstructed trajectories $\{\tau'_1, ..., \tau'_N\}$

- **Filtering:** We now need to evaluate the collected reconstructed trajectories $\{\tau'_1, ..., \tau'_N\}$ and determine whether to accept or reject $\tau' \in \{\tau'_1, ..., \tau'_N\}$. Specifically, we accept $\tau'$ as follows:

$$A(\tau, \tau') = 1_{R(\tau')>R(\tau)} \tag{18}$$

This means we greedily filter out the candidates $\{\tau'_{1'} ... \tau'_{N'}\} \subset \{\tau'_1, ..., \tau'_N\}$ that have a higher reward than $\tau_m$, which has a higher possibility of reaching the goal. Then we return these trajectories and add them into the replay buffer $\mathcal{D}$.

# F    CASE STUDY

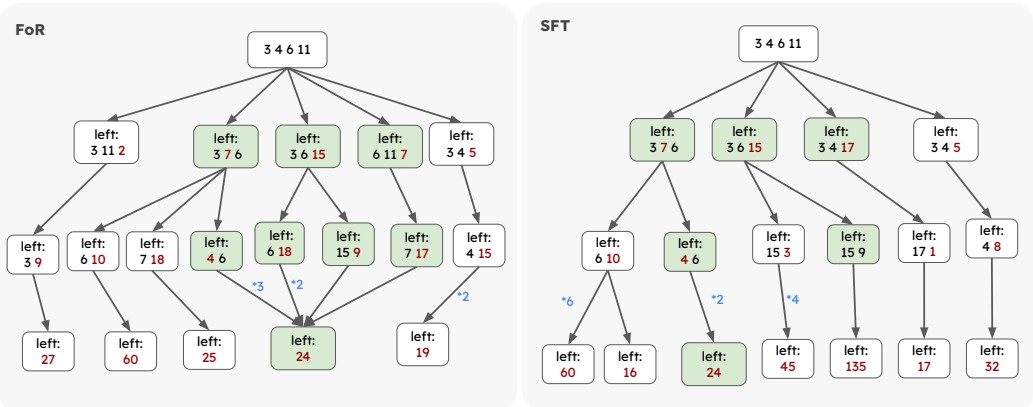

Figure 9: Problem *(3,4,6,11)*. Green blocks represent the states that can achieve 24. Blue numbers represent the sample times of trajectories bigger than 1. This shows that FoR can achieve 24 in multiple different states while SFT usually fails to do so.

**Balance between diversity and accuracy.**    According to Figure 9, we use the problem *(3,4,6,11)* to show how FoR achieves such high performance while focusing on diversity. As illustrated in the figure, we compare trajectories sampled 20 times by both SFT and FoR. While both methods produce diverse trajectories initially, FoR demonstrates better capability in reaching successful final steps from various middle steps. For example, FoR successfully transitions from intermediate steps *(3,6,15)* to target 24, whereas SFT fails to do so. This highlights the effectiveness of FoR's design in simultaneously promoting diversity and ensuring accuracy.

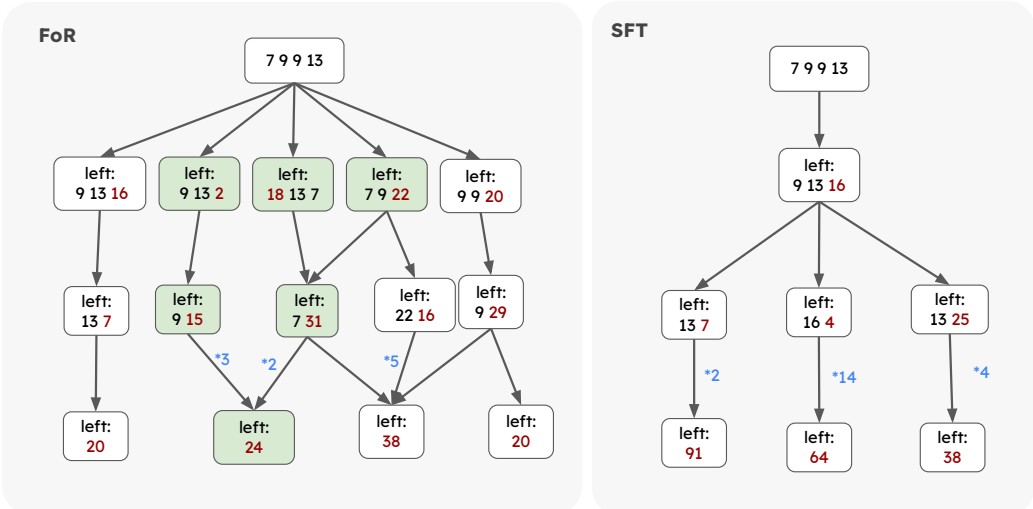

Figure 10: Problem *(7,9,9,13)*. Green blocks represent the states that can achieve 24. Blue numbers represent the sample times of trajectories bigger than 1. This shows that FoR can achieve 24 in multiple different states while SFT usually fails to do so.

**Better robustness due to exploratory nature.** According to Figure 10, we use the problem *(7,9,9,13)* to demonstrate the robustness of FoR. As shown in the figure, SFT repeatedly fails by getting stuck in a single second state of *(9,3,16)* 20 times, while FoR successfully discovers multiple diverse trajectories leading to the correct solution. This robustness can be attributed to the exploratory nature of FoR's training objective, which encourages the model to sample diverse successful trajectories. By expanding the search space through high-reward exploration, FoR increases the chance of finding successful outcomes. This capability not only improves the robustness of the model but also enhances its generalization to new scenarios, showcasing the effectiveness of FoR in addressing complex reasoning tasks.

**Other examples** Figure 11 shows generated samples for the BlocksWorld, Figure 12 for Game24, and Table 12 for PrOntoQA, respectively.

Table 12: Examples for PrOntoQA.

| State | Action |
|---|---|
| **Query:** True or false: 31 is not imaginary. (OOD) | |
| 31 is a natural number. | Natural numbers are integers. |
| 31 is an integer. | Integers are real numbers. |
| 31 is a real number. | Real numbers are not imaginary. |
| 31 is not imaginary. | Finish. |
| **Query:** True or false: Wren is not bony. (In-distribution) | |
| Wren is a painted lady. | Each painted lady is a butterfly. |
| Wren is a butterfly. | Each butterfly is a lepidopteran. |
| Wren is a lepidopteran. | Each lepidopteran is an insect. |
| Wren is an insect. | Each insect is an arthropod. |
| Wren is an arthropod. | Each arthropod is not bony. |
| Wren is not bony. | Finish. |

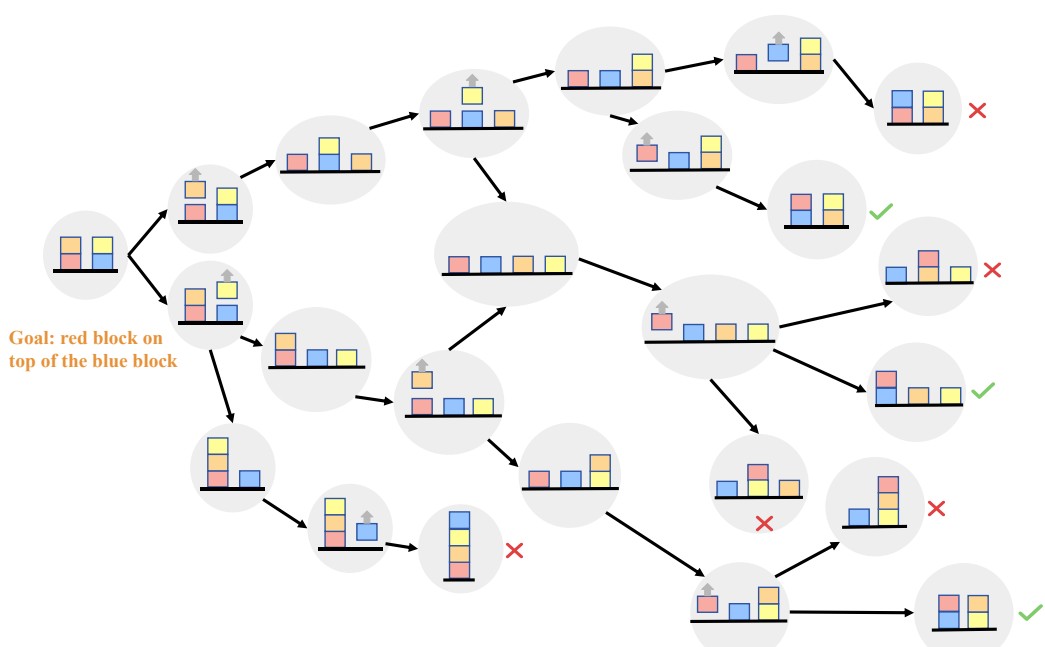

Figure 11: Example of BlocksWorld for 6-step planning.

## G    LIMITATIONS AND FUTURE WORK

Due to resource constraints, our experiments use language models with up to 13B parameters. However, we expect FOR to hold for larger models as well, and may potentially benefit larger models even more. Recent works (Xi et al., 2024; AlKhamissi et al., 2023) that finetune larger models to improve their reasoning ability with maximizing objectives usually need a large amount of data, and our data-efficient FOR may improve this process. Future work should further address two limitations in FOR.

The first is **aquisation of a large amount of trajectories efficiently**. Online sampling with LLMs is computationally expensive, leading to more efficient and effective strategies for exploring more complex settings such as real-world settings AlfWorld (Shridhar et al., 2020) and TravelPlanner (Xie et al., 2024a) to be further studied.

The second is **faciliating FOR long-range steps reasoning.** LLMs fall short in long-range planning and reasoning, thus methods like MCTS (Feng et al., 2023) or automatic reasoning system (Trinh et al., 2024) can be combined with LLMs for long-horizon diverse reasoning.

## H    BROADER IMPACT

This study introduces FOR, a methodology that trains LLMs as policies to solve complex reasoning problems with better creativity and diversity. We believe this work connects LLM reasoning with GFlowNets and contributes to the application of GFlowNets to LLMs.

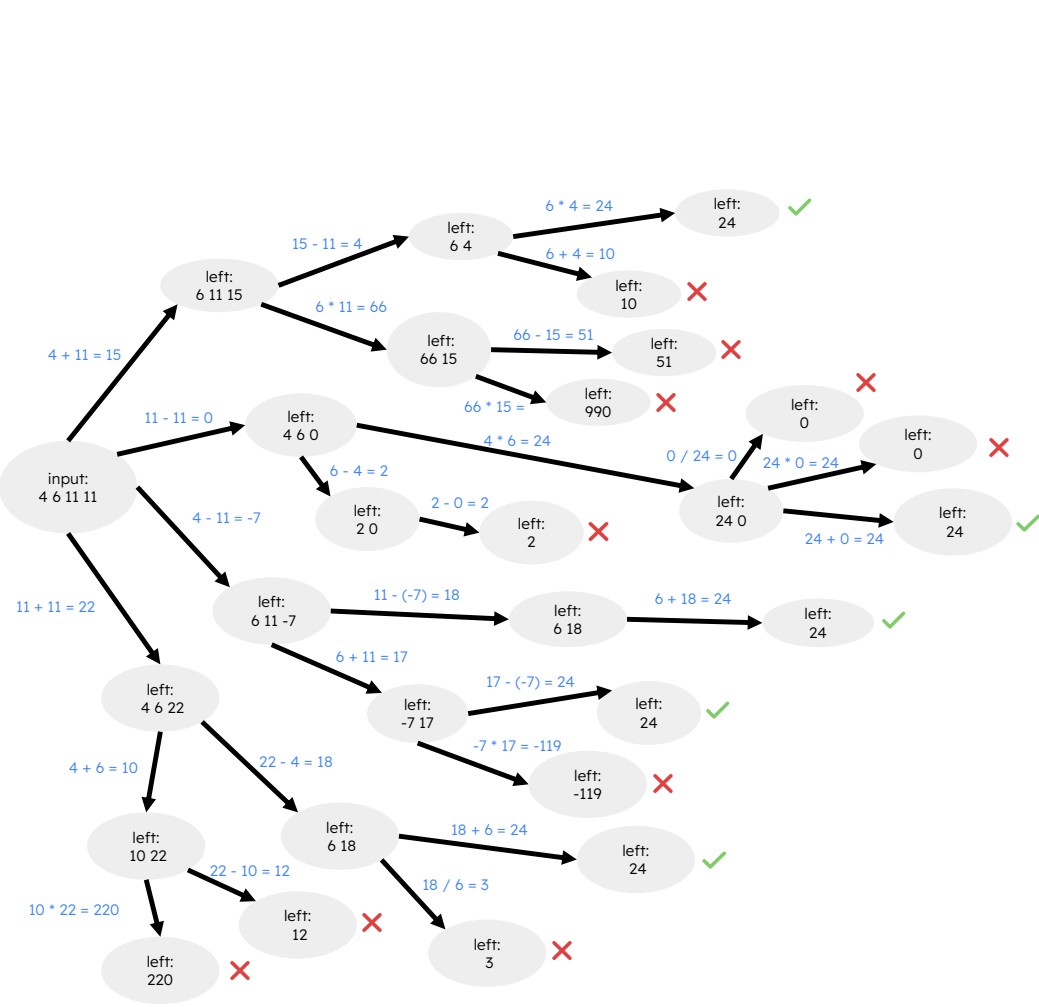

Figure 12: Example of game of 24.

