# OpenReview forum: "Flow of Reasoning: Training LLMs for Divergent Problem Solving with Minimal Examples"
_ICLR.cc/2025/Conference — Submitted to ICLR 2025_

### Official Review · Reviewer_iYFq · 2024-11-01

**Soundness:** 3
**Presentation:** 3
**Contribution:** 3
**Rating:** 5
**Confidence:** 3

**Summary:**

The paper introduces FOR, a novel method designed to enhance the divergent reasoning capabilities of LLMs. FOR addresses the challenge of generating diverse solutions to complex problems by formulating multi-step reasoning as a Markovian flow on a DAG structured reasoning graph. This approach allows the incorporation of GFlowNet techniques to train LLMs to sample diverse and high-quality reasoning paths proportional to the unnormalized reward of target problems. The paper demonstrates FOR's effectiveness across five diverse reasoning tasks, showing significant improvements in discovering creative and high-quality solutions with minimal training data compared to existing methods.

**Strengths:**

- well-written and easy to follow
- The idea of GFlowNet is quite interesting

**Weaknesses:**

I believe this is an interesting piece of work with a highly valuable topic. However, I think the benchmarks considered in this paper, such as Game24, PrOntoQA, Rubik's Cube, etc., do not actually require much divergent thinking. I strongly recommend directly considering some works on lateral thinking [1][2][3] to assess whether the proposed methods are truly effective. I understand that it might not be possible to fully attempt these benchmarks during the rebuttal stage, but it could be possible to try some examples or add a related works section for lateral thinking or creativity.



[1] Let’s Think Outside the Box: Exploring Leap-of-Thought in LLMs with Creative Humor Generation, CVPR24

[2] Lateral Thinking Puzzles for Large Language Models, EMNLP23

[3] RiddleSense: Reasoning about Riddle Questions Featuring Linguistic Creativity and Commonsense Knowledge, ACL21

**Questions:**

see weakness

---

> ### Author Response · Authors · 2024-11-21
> **Response to Reviewer iYFq**
>
> Thank you for the positive and insightful comments!
>
> ## Response to Weaknesses
> >W1: The benchmarks in this paper do not require much divergent thinking. Consider lateral thinking.
>
> **Response-W1:**
> Thank you for your thoughtful feedback and suggestions. We address your concerns and provide additional insights into our work below.
>
> ### Concern about Unsuitable Benchmarks
> Firstly, we respectfully argue that benchmarks like Game24 and Rubik's Cube are indeed suitable for evaluating the diversity of our results, as they often involve multiple trajectories to achieve a single goal. For example, as shown in Figure 1 of our paper, both Game24 and Rubik's Cube exhibit diverse solution paths, making them effective benchmarks for assessing the exploratory nature of our method. Additionally, although PrOntoQA is not allowed to have a diverse solution, it (and other benchmarks) still benefits from our method’s ability to avoid getting stuck in incorrect trajectories (please refer to the **Qualitative Results** in the **General Response**). Specifically, FoR encourages effective exploration in the search space during inference with repeated samplings, which increases the likelihood of finding correct and diverse trajectories.
>
> ---
>
> ### Additional GSM8K Experiments
> To further address your suggestion, we conducted additional experiments on GSM8k math problems which are deemed more open-ended than the tasks we've done. Following RAP[4], we define an action as an intermediate sub-question to solve the problem and a state as the history of all intermediate sub-question-answer pairs.
>
> We perform experiments in a 2-shot setting, comparing FoR with Supervised Fine-Tuning (SFT), CoT, CoT with self-consistency (CoT-SC), and RAP [4]. For each problem, we sample 4 solutions, considering success as achieving at least one correct solution when calculating accuracy. For training, we use the last 50 examples from the GSM8k dataset. The baselines are implemented based on [4]. Since no established metrics exist for evaluating the diversity of open-ended mathematical reasoning, we manually annotated 50 test examples that have multiple reasoning trajectories to assess the similarity between reasoning trajectories, determining whether two reasoning paths were semantically equivalent or not.
>
> As shown in the table below, the results demonstrate that FoR is effective on GSM8k, achieving higher accuracy than all baselines while maintaining the diversity of reasoning trajectories. These findings further validate the generalizability of FoR across different reasoning benchmarks.
>
>
> | Method     | Acc. (%) | Diversity |
> |------------|---------|--------------|
> | CoT-SC     | 41.74   | -            |
> | CoT        | 45.72   | 1.12           |
> | RAP        | 37.16   | -            |
> | SFT        | 52.69   | 1.13           |
> | FoR (Ours) | **57.39**   | **1.26**   |
>
> ---
>
> ### Discussion of Suggested Works
> We appreciate your recommendation to consider the works in lateral thinking [1, 2, 3].
>
> Upon review, we think that:
> * [1] investigates humor generation, emphasizing the ability to interpret figures or questions with multiple plausible meanings.
> * [2] and [3] focus on riddle-solving QA tasks that require reasoning about unexpected or unconventional answers.
>
>
> While these works emphasize lateral thinking—focusing on looking at problems from *new perspectives and defying preconceptions* associated with the *right-brain hemisphere* [2,3]—our work aligns with vertical thinking, which involves *logical, sequential reasoning* typically linked to the *left-brain hemisphere* [2,5]. Our approach emphasizes exploring diverse solutions within a structured, logical framework rather than breaking away from it. Unlike lateral thinking, which defines creativity as the ability to defy conventional frameworks and generate surprising content, our work defines creativity as the diversity of reasoning paths within the context of multi-step problem-solving tasks (e.g., math) with a clearly defined and known goal state. Therefore, we believe these works are **orthogonal** to ours, as they address fundamentally different aspects of creativity and reasoning.
>
> ---
> References
>
> [1] Let’s Think Outside the Box: Exploring Leap-of-Thought in LLMs with Creative Humor Generation, CVPR24
>
> [2] Lateral Thinking Puzzles for Large Language Models, EMNLP23
>
> [3] RiddleSense: Reasoning about Riddle Questions Featuring Linguistic Creativity and Commonsense Knowledge, ACL21
>
> [4] Shibo Hao, et al. Reasoning with Language Model is Planning with World Model. In Proceedings of the 2023 Conference on Empirical Methods in Natural Language Processing, pages 8154–8173, Singapore. Association for Computational Linguistics.
>
> [5] Waks, S. Lateral Thinking and Technology Education. Journal of Science Education and Technology 6, 245–255 (1997).

---

> ### Author Response · Authors · 2024-11-25
>
> Dear Reviewer,
>
> This is a gentle reminder that the discussion period is closing soon. We've responded to all concerns, providing detailed clarifications and new experimental results. Please feel free to let us know if you have any further questions. We'd greatly appreciate it if you could consider raising scores and/or engaging in further discussions. Thanks!

---

> ### Author Response · Authors · 2024-11-25
> **Follow-Up on Score Change by Authors**
>
> Dear Reviewer,
>
> We noticed that you reduced your score from 6 to 5, and we’d greatly appreciate it if you could share any additional concerns or feedback that caused the change. We have responded to your initial questions in the original review and provided new experimental results on more "open-ended" problems (e.g., GSM8K). If there are any new issues you’d like us to address, we’d be happy to do so.
>
> Thank you for your time and consideration!

---

> ### Author Response · Authors · 2024-11-27
> **Added discussion into revised paper**
>
> Dear Reviewer,
>
> We’d like to thank you again for your suggestion on the related work. In our updated PDF, we have cited and added more discussion of the suggested work. Due to the time and space limit, it's challenging for us to make this change to the main paper now. We currently included it in the section of “Lateral Thinking and Vertical Thinking” in Appendix A, and we will make sure to move to the main paper in the camera-ready version. We look forward to more discussion!

---

> ### Author Response · Authors · 2024-12-02
> **Gentle Reminder**
>
> Dear Reviewer,
> This is a gentle reminder that the discussion period will be closing soon. We have addressed all concerns by providing detailed clarifications and updated experimental results. Please don't hesitate to reach out if you have any additional questions. We would greatly appreciate it if you could consider raising the scores and/or engaging in further discussions. Thank you!

---

### Official Review · Reviewer_h3sb · 2024-11-04

**Soundness:** 2
**Presentation:** 2
**Contribution:** 2
**Rating:** 5
**Confidence:** 4

**Summary:**

This paper proposes Flow of Reasoning (FOR), a data-efficient finetuning method that enables large language models (LLMs) to generate diverse, high-quality solutions for multi-step reasoning tasks. Unlike existing methods that prioritize accuracy or highest-reward solutions, FOR promotes solution diversity by modeling reasoning as a Markovian flow on a directed acyclic graph (DAG), applying principles from Generative Flow Networks (GFlowNets) to sample multiple reasoning paths based on reward. With minimal data requirements (around 15 examples), FOR significantly outperforms traditional approaches across five complex puzzle-solving tasks, demonstrating its effectiveness in improving both reasoning quality and diversity.

**Strengths:**

The paper effectively addresses a gap in existing LLM approaches by focusing on solution diversity in multi-step reasoning tasks, which enhances robustness and creativity in applications like scientific discovery.

The FOR framework introduces a unique approach by modeling multi-step reasoning as a Markovian flow on a directed acyclic graph (DAG), using Generative Flow Networks (GFlowNets) principles. This higher-level, step-based approach enables more efficient sampling of diverse reasoning paths, distinguishing FOR from traditional token-level models.

FOR demonstrates impressive data efficiency, requiring only about 15 training examples to achieve competitive results. This efficiency makes the method more feasible for practical applications where extensive labeled data is limited.

Through comprehensive testing across five complex tasks (embodied reasoning, logical reasoning, spatial reasoning, etc.), FOR outperforms multiple baselines by 20%–85%, showing versatility and robustness across different reasoning challenges.

**Weaknesses:**

The primary issue I find with this paper is the lack of clarity regarding any original insights or core observations the authors provide about large language models in the context of reasoning with diversity. Throughout the paper, it seems that all key observations, points, and techniques are primarily derived from existing works, without presenting novel contributions from the authors. For example, the authors note that existing LLMs focus on generating the reasoning path with the highest reward. However, LLMs inherently exhibit randomness and uncertainty in their reasoning processes, which could naturally encourage exploration across diverse reasoning paths. In additional papers, such as TR [1] and CR [2], they all build a DAG toward getting solutions.

The second issue is that there are plenty of high-level ideas and "big words" in the explanation parts of the paper. For instance, in Line 172, what does "As discussed in §1, to generate diverse, high-quality reasoning trajectories for solving a task, we want to sample the trajectories with probabilities proportional to the reward. " Why sampling in such a way can generate not only diverse but also high-quality reasoning paths? What is the logic between this strong conclusion and its previous and subsequent sentences?

The third issue is that this paper introduces too many existing methods, ideas, equations, and conclusions from prior works. However, the way the authors organize these elements does not integrate them effectively, resulting in a lack of clear and logical explanations to support an understanding of their own contribution. While reading the paper, I found the frequent citations, numerous "See appendix" references, and lack of important insights to be distracting and disruptive to my understanding. For example, the authors do not provide clear or consistent notation descriptions. It appears that notations are created on the spot whenever needed, leading to confusion and a lack of coherence throughout the paper.

The fourth issue is that some of the questions posed in the paper are unclear, leading to confusion about their intent and relevance. For example, (1), in Line 188, why the distribution directly equals to F(\tau)/Z? To me, this should be an approximation.; (2), in Line 191, where is the P(s_0) that is the prior in this step-wise distribution? (3). in Line 202, when F is defined for the state flow, it's unclear why F(s_n) is to be equal to R.

The final issue is that the experiment section does not provide sufficient persuasiveness for the paper. For example, if the goal of this paper is to fine-tune the LLM to generate diverse reasoning paths, the main content should provide a more specific and detailed explanation of how to fine-tune LLMs according to the proposed objectives. Besides, how the authors define diversity is unclear to me. They may need to count successful trajectories a policy finds for the successful example. But how the two trajectories can be compared when there are many ones? Thus, during training, how can you ensure that the LLM can generate two different paths from one state? How about comparing your methods with other similar (using DAG) methods [1], [2]?

I was not impressed by the optimal results in the table due to the lack of sufficient explanation, in-depth analysis, and insights in the paper.

[1]. Chen, et.al., Toward Adaptive Reasoning in Large Language Models with Thought Rollback, ICML24.

[2]. Zhang, Cumulative reasoning with large language models, ICLR Workshop24.

**Questions:**

See Weaknesses Above.

---

> ### Author Response · Authors · 2024-11-21
> **Response to Reviewer h3sb - Part 1**
>
> Thank you very much for your insightful comments and suggestions! Below, we address each of comments in detail.
>
> ## Response to Weaknesses
>
> >W1.1: lack of clarity regarding any original insights or core observations the authors provide about large language models in the context of reasoning with diversity. Throughout the paper, it seems that all key observations, points, and techniques are primarily derived from existing works, without presenting novel contributions from the authors.
>
> **Response-W1.1:**
>
> Please refer to the **Qualitative Results** in the **General Response** to see the insight of our methods. Also, please refer to the contributions part in the **General Response**.
>
> ---
>
>
> >W1.2: LLMs inherently exhibit randomness and uncertainty in their reasoning processes, which could naturally encourage exploration across diverse reasoning paths.
>
> **Response-W1.2:**
>
> We would like to clarify that, though LLMs exhibit randomness during generation (e.g., with random sample decoding), it **does not** lead to diverse and high-quality reasoning paths. Indeed, our experiments have done extensive study of LLMs with various diversity-enhancing decoding strategies, such as Nucleus Sampling, Typical Sampling, and diverse beam search (Line.287-290). Results show these common methods fail to generate diverse and accurate solutions at the same level as our proposed method (see Table.1). Our proposed approach instead generates solutions with probabilities proportional to the solutions' "goodness" (i.e., reward). This mechanism produces diverse and high-quality solutions.
>
> ---
>
> >W1.3: In additional papers, such as TR [1] and CR [2], they all build a DAG toward getting solutions.
>
> **Response-W1.3:**
>
> Thank you for pointing out these works. We'd like to clarify that the works TR and CR that build DAGs still aim to find the max-reward solution instead of diverse high-quality solutions. Thus these works are similar to the methods we've compared with, such as RAP and ToT. Our approach achieves better accuaracy and diversity (e.g., Table.1). In addition, the TR and CR methods use revisions and multiple rounds of prompting, which are orthogonal and complementary to our method.
>
> ---
>
>
> >W2: Why sampling in such a way can generate not only diverse but also high-quality reasoning paths? What is the logic between this strong conclusion and its previous and subsequent sentences?
>
> **Response-W2:**
>
> Sampling in proportion to reward, as used in our method, facilitates the generation of both diverse and high-quality reasoning paths by aligning the sampling process with the reward distribution [2, 3]. This mechansim encourages to discover multiple solutions, each evaluated and sampled based on its "goodness" as determined by the reward. The exploration for diverse solutions also makes the model more robust and generalize better. A concrete example is analyzed in the **Qualitative Results** in the **General Response**: traditional methods like SFT fails to explore alternative reasoning paths, becoming *stuck* in specific incorrect trajectories. In contrast, our method explores and uncover alternative valid solutions with high rewards.
>
>
> ---
>
> >W3: This paper introduces too many existing methods, ideas, equations, and conclusions from prior works. However, the way the authors organize these elements does not integrate them effectively, resulting in a lack of clear and logical explanations to support an understanding of their own contribution.
>
> **Response-W3:**
>
> Please refer to the **Contribution** part in the **General Response** for a summary of our contributions. In particular, we're the first to formulate multi-step LLM reasoning as Markovian flow. This formulation enables us to adapt and integrate existing successful approaches (e.g., GFlowNets objectives, local search strategies) to our new setting of multi-step reasoning. We believe the abilility to generalize and integrate existing successful approaches (instead of having to re-invent all technical components from scratch) is a key advantage of our framework. We will make this more clear in the revised paper.

---

> ### Author Response · Authors · 2024-11-21
> **Response to Reviewer h3sb - Part 2**
>
> >W4: Some of the questions posed in the paper are unclear.
>
> **Response-W4:**
>
> **Q4.1:** "*in Line 188, why the distribution directly equals to $F(\tau)/Z$*?"
>
> **A4.1:** This follows directly the definition of a normalized probability distribution in flow-based models (e.g., Equation (3) in [1]). More specifically, the reason the distribution equals $\frac{F(\tau)}{Z}$ is that $F(\tau)$ represents the flow associated with trajectory $\tau$, and $Z$ is the total flow (partition function), which normalizes $F(\tau)$ over all possible trajectories. This normalization ensures that the resulting value is a valid probability distribution, as probabilities must sum to 1 across all trajectories.
>
> Intuitively, think of  $F(\tau)$  as assigning a "weight" or "importance" to each trajectory  $\tau$. Dividing by $Z$, which is the sum of these weights across all trajectories, converts these weights into probabilities that reflect the relative likelihood of each trajectory.
>
> **Q4.2** "*in Line 191, where is the $P(s_0)$ that is the prior in this step-wise distribution?*"
>
> **A4.2:** $P(s_0)=1$. This is because, by the definition of the Markovian flow, all trajectories start with the same initial state $s_0$ [2]. That is, the probability of a trajectory passing through the initial state $s_0$ is 1. We will make this clearer in the revised version.
>
> **Q4.3:** "*in Line 202, when F is defined for the state flow, it's unclear why $F(s_n)$ is to be equal to R.*"
>
> **A4.3:** As explained in the response to W4, the diversity-promoting reasoning is achieved by sampling reasoning trajectories with probabilities proportional to the reward. By definition of state flow, $F(s_n)$ represents the cumulative flow reaching the terminal state $s_n$. The equality $F(s_n) = R$ ensures that the terminal state flows are directly aligned with the reward distribution. This alignment ensures that the trajectories leading to $s_n$ are sampled in proportion to their contribution to $R$.
>
>
>
> ---
>
>
> >W5: The experiment section does not provide sufficient persuasiveness for the paper.
>
> **Response-W5:**
>
> **Q5.1:** "*The main content should provide a more specific and detailed explanation of how to fine-tune LLMs according to the proposed objectives.*"
>
> **A5.1:** We’ve described the fine-tuning objective and efficient approximation in Section 3.2.  The experiment section describes specific instantiations of the fine-tuning components in each task, such as the reward design. Appendix D further provides the detailed algorithm and all prompts used in fine-tuning to elaborate on the finetuning procedure. We’ll make the details clearer.
>
> **Q5.2:** "*How the authors define diversity is unclear to me.*"
>
> **A5.2:** Thank you for your feedback. We report the average number of unique correct solutions among the $n$ (e.g., 20) sampled trajectories (Line.302). This means that for each problem, we calculate the number of different correct trajectories out of $n$ samples. We then compute the average of these values across the test set, which serves as our Diversity metric. We have also included more detailed definition of the diversity metric in Appendix C.1. We will ensure this explanation is clearer in the next revision.
>
> **Q5.3:** "*But how the two trajectories can be compared when there are many ones?*"
>
> **A5.3:** We use exact string match to determine uniqueness between two trajectories. This comparison works because the output of our tasks are in a certain standardized format.
>
> **Q5.4:** "*Thus, during training, how can you ensure that the LLM can generate two different paths from one state?*"
>
> **A5.4:** During training, we encourage the model to generate different trajectories in a "soft" way, by setting a high temperature ($\alpha=1$) that increases randomness of LLM generation, as well as using local search (Section.3.2.2) which further promotes exploration of different trajectories. Crucially, the training objective uses reward to give feedback to these trajectories, to make sure the model learns to generate not only diverse but also high-quality trajectories.
>
> ---
> References
>
> [1]. Malkin, Nikolay, et al. "Trajectory balance: Improved credit assignment in gflownets." Advances in Neural Information Processing Systems 35 (2022)
>
> [2] Bengio, Yoshua, et al. "Gflownet foundations." The Journal of Machine Learning Research 24.1 (2023)
>
> [3] [The GFlowNet Tutorial](https://milayb.notion.site/The-GFlowNet-Tutorial-95434ef0e2d94c24aab90e69b30be9b3)

---

> ### Author Response · Authors · 2024-11-25
>
> Dear Reviewer,
>
> This is a gentle reminder that the discussion period is closing soon. We've responded to all concerns, providing detailed clarifications and new experimental results. Please feel free to let us know if you have any further questions. We'd greatly appreciate it if you could consider raising scores and/or engaging in further discussions. Thanks!

---

> ### Author Response · Authors · 2024-12-02
> **Gentle Reminder**
>
> Dear Reviewer,
> This is a gentle reminder that the discussion period will be closing soon. We have addressed all concerns by providing detailed clarifications and updated experimental results. Please don't hesitate to reach out if you have any additional questions. We would greatly appreciate it if you could consider raising the scores and/or engaging in further discussions. Thank you!

---

> ### Comment · Reviewer_h3sb · 2024-12-02
> **Acknowledgments to the Author's Reply**
>
> Thank you for answering my questions!
>
> I am pleased to see that some of my concerns have been acknowledged and solved, and I hope the authors will continue to polish and improve the paper as indicated in their response.
>
> However, I personally believe this submission has some weaknesses, leading me to conclude that a score of 5 is fair. Some of my concerns and identified weaknesses can be found in the 'Further Comments' section above. Here is the final conclusion paragraph as a quick snapshot:
>
> "In summary, I hold my view that this work, which claims too many firsts and breakthroughs but lacks the necessary definitions and discussions of related work, uses too much content from existing work (formulas, definitions, mathematical explanations), and directly presents the best numerical results but lacks convincing discussions and insights, does not reach the borderline ICLR. Therefore, I believe that 'marginally below the acceptance threshold' is a fair assessment, particularly after thoroughly reviewing this submission multiple times."
>
> Therefore, I am relatively confident that 'marginally below the acceptance threshold' is my final assessment after thoroughly reading the paper multiple times and engaging in discussions with the authors.
>
> Thank you very much!

---

> ### Author Response · Authors · 2024-12-04
> **Response to Reviewer h3sb**
>
> Thank you for your comments. We believe our previous responses have addressed your questions directly. Here we’d like to re-emphasize key points from our previous replies. We’d appreciate any further comments *on the basis of our previous responses* to make progress in the discussion.
>
> - Our paper **has already cited, discussed, and empirically compared** existing works on LLM generation diversity works, such as [1], as well as many decoding methods including sampling with varying temperatures, Nucleus Decoding, Typical Decoding, and Diverse Beam-Search [2,3,4] (See Table.1). As discussed in several places in the paper (e.g., Line.92 and Line.144), these methods (including the four papers you mentioned) use ***token-level*** modeling of LLM generation. However, such token-level approaches are less effective for handling complex multi-step reasoning tasks that require generating reasoning trajectories consisting of multiple steps (each comprising multiple tokens). Our work introduces higher-level modeling at the ***granularity of reasoning steps*** for better promoting diversity in this multi-step reasoning setting. As discussed in Introduction section (Line.95), this effectively combines the best of the previous token-level diversity-seeking approaches (particularly GFlowNets) and the recent multi-step reasoning methods such as Tree-of-Thoughts and RAP (which search at the step-level for max-reward trajectory), while overcoming their respective limitations.
>
> Therefore, as clarified earlier, our focus is on promoting diversity in LLM multi-step reasoning by designing a novel approach tailored to this specific setting. While general token-level generation can be interpreted as "*inherently performing multi-step reasoning*," this perspective overlooks the potential for developing more effective methods specifically suited to the multi-step reasoning setting.
>
> - Thanks for acknowledging MCTS is not equivalent to Markovian Flow. We’ve summarized our "novelty" in the General Response. As again clarified in our previous response, adapting the *classical* Markovian Flow formalism to the *new* setting of multi-step LLM reasoning opens the door for us to connect/adapt successful existing approaches (proposed in other settings) to address this significant new problem effectively. We’d like to highlight that **the ability to re-purpose and generalize successful existing approaches to new problem setting (instead of having to re-invent all details from scratch) is a key advantage** of our work. We’d be happy to discuss if there are different perspectives on the expectations of "novelty".
>
> - We’ve explained, in both the initial and follow-up responses, the “insights” of why our approach improves both accuracy and diversity. We’ve explained the intuition behind our method and also provided a qualitative example (in **General Response**) to illustrate this (i.e., the exploratory nature of FoR encourages diverse explorations and helps avoid the reasoning getting stuck in bad steps, thus improving accuracy while producing diverse solutions). We’d appreciate more discussion on the basis of our responses.
>
> [1] Hu, Edward J., et al. "Amortizing intractable inference in large language models." International Conference on Learning Representations (2024) Oral.
>
> [2] Holtzman, A., et al. "The Curious Case of Neural Text Degeneration." International Conference on Learning Representations. (2020)
>
> [3] Meister, C., et al. 2023. "Locally Typical Sampling" Transactions of the Association for Computational Linguistics. (2023)
>
> [4]Vijayakumar et al., "Diverse Beam Search: Decoding Diverse Solutions from Neural Sequence Models" AAAI (2018)

---

### Official Review · Reviewer_3ETe · 2024-11-04

**Soundness:** 3
**Presentation:** 3
**Contribution:** 2
**Rating:** 5
**Confidence:** 3

**Summary:**

Paper proposed a new approach for generating diverse reasoning traces using GFlowNets.

**Strengths:**

- Paper is organized cleanly and easy to understanding
- Experiments involve a variety of tasks and baselines
- Approach shows improvements to accuracy and diversity of model generations

**Weaknesses:**

Although a variety of tasks were evaluated, all of the tasks are relatively structured and uses constrained action spaced (such as STACK, UNSTACK, PUT, PICKUP in BlocksWorld). In contrast, reasoning tasks that are useful to real users are typically much more open ended, with a more diverse variety of both questions and potential answer formats. In order to show that the proposed approach is scalable to these more open ended reasoning tasks, it would be great to also evaluate on benchmarks like MATH or GSM8k.

The proposed approach requires a reward function, which, in the experiments, required different kinds of reward shaping for each task. Designing specialized reward functions for each task is not practical when training general-purpose LLMs on a wide range of different tasks. It would be great to either show that the proposed approach leads to improvements in accuracy and diversity with just the sparse "success" reward, or use a shaped reward that can work with all tasks.

It would be great to show qualitative examples of the different responses generated by each approach, and highlight how they differ from standard approaches

**Questions:**

See weaknesses

---

> ### Author Response · Authors · 2024-11-21
> **Response to Reviewer 3ETe**
>
> Thank you very much for your insightful comments and suggestions! We address each of comments in detail below.
>
>
> ## Response to Weaknesses
>
> ---
>
>
> >W1：Although a variety of tasks were evaluated, all of the tasks are relatively structured and uses constrained action spaced (such as STACK, UNSTACK, PUT, PICKUP in BlocksWorld). In contrast, reasoning tasks that are useful to real users are typically much more open ended, with a more diverse variety of both questions and potential answer formats. In order to show that the proposed approach is scalable to these more open ended reasoning tasks, it would be great to also evaluate on benchmarks like MATH or GSM8k.
>
> **Response-W1:**
>
> Thanks for your suggestion. We add an additional experiment of GSM8K to evaluate the performance of our method on a open-ended reasoning task. We follow [1] to define an action as an intermediate sub-question to solve the problem and a state as all the history intermediate pairs of a sub-question and its answer.
>
> We conduct experiments with 2-shot settings, and compare them with supervised fine-tuning (SFT), CoT, CoT with self consistency (CoT-SC), and RAP. For each problem, we sample 4 solutions and the success is indicated as long as 1 solution is correct. For training, we construct the training dataset with the last 50 examples in the GSM8K training set. The implementation of baselines refers to [2]. Due to the lack of established evaluation metrics for assessing the diversity of open-ended mathematical reasoning, we manually annotate 50 test examples to evaluate the similarity between reasoning trajectories, determining whether two reasoning trajectories are semantically equivalent or not.
>
> The results are shown below: FoR shows effectiveness on GSM8k and exceeds the accuracy and diversity of all baselines, which demonstrates the potential of FoR for extending to more open-ended reasoning tasks.
>
> As shown in the table below, in the task of GSM8k, FoR outperforms all baselines in both accuracy (by an absolute improvement of 4.7% over SFT) and diversity (by an absolute improvement of 0.13 over SFT). These results highlight the potential of FoR for extending to more open-ended reasoning tasks.
>
> | Method     | Acc.(%) | Diversity |
> |------------|---------|--------------|
> | CoT-SC     | 41.74   | -            |
> | CoT        | 45.72   | 1.12           |
> | RAP        | 37.16   | -            |
> | SFT ($\alpha$=1.0)       | 52.69   | 1.13           |
> | FoR (Ours) | **57.39**   | **1.26**   |
>
> ---
>
> >W2: Use a personalized reward function for each task. Not practical when training general-purpose LLMs on a wide range of different tasks.
>
> **Response-W2:**
>
> We would like to clarify that the reward functions used in our paper do not heavily rely on human design. Specifically, they consist of a standard success-or-failure reward and a value that that measures the goodness of an action to achieve a goal, which is provided by an additional LLM or Environment (Line.351, Line.450-451). This reward formulation is consistent with the previous RL framework for LLM reasoning (e.g., RAP [1]).
>
>
> We would like to note that in the additional GSM8k experiment, we only use a standard success-or-failure reward during training, which is also effective, as demonstrated in the response to W1. Overall, our approach is applicable to a wide range of tasks as long as a reasonable and reliable reward function/model is available, similar to the common RL approaches [3] applied to LLM reasoning .
>
>
> ---
>
> > W3: Great to show qualitative examples of the different responses generated by each approach, and highlight how they differ from standard approaches
>
> **Response-W3:**
>
> Please refer to the **Qualitative Results** in the **General Response**.
>
> ---
>
> References
>
> [1] Hao, Shibo, et al. "Reasoning with Language Model is Planning with World Model." Proceedings of the 2023 Conference on Empirical Methods in Natural Language Processing. 2023.
>
> [2] Hao, Shibo, et al. "LLM Reasoners: New Evaluation, Library, and Analysis of Step-by-Step Reasoning with Large Language Models." First Conference on Language Modeling.
>
> [3] Wang, C., Deng, Y., Lyu, Z., Zeng, L., He, J., Yan, S., & An, B. (2024). Q*: Improving multi-step reasoning for llms with deliberative planning. arXiv preprint arXiv:2406.14283.

---

> ### Author Response · Authors · 2024-11-25
>
> Dear Reviewer,
>
> This is a gentle reminder that the discussion period is closing soon. We've responded to all concerns, providing detailed clarifications and new experimental results. Please feel free to let us know if you have any further questions. We'd greatly appreciate it if you could consider raising scores and/or engaging in further discussions. Thanks!

---

> > ### Comment · Reviewer_3ETe · 2024-11-25
> >
> > Thanks to the authors for rebuttal, and apologies for the delayed response.
> >
> > Regarding the new GSM8k experiments, I am wondering why you train on only the last 50 train examples, rather than the full dataset?
> >
> > Additionally, you mentioned that for the evaluation, you "sample 4 solutions and the success is indicated as long as 1 solution is correct". This metric can be improved or worsened by simply adjusting the temperature when sampling. I am wondering whether you can perform a more thorough evaluation, e.g. sweeping temperatures of (2, 1, 0.5, 0), for the FoR and SFT methods, and logging both the average and best-of-N accuracies? It would be interesting to have a better understanding of how training with the proposed approach compares to increasing the temperature for generating diverse samples. No need to evaluate the diversity measure, since I understand that it is more challenging to evaluate.

---

> > > ### Author Response · Authors · 2024-11-27
> > >
> > > Thank you for your feedback and for taking the time to engage with our rebuttal. We appreciate your suggestions for improving the evaluation.
> > >
> > > **Response to Q1:** The reason why we use 50 training data is we would like to evaluate our method under low-data scenarios, which is one of key focuses in our work. This aligns with other experimental settings used in our paper (e.g., in Section 4.6 Logical Reasoning, we also used 50 training examples), where we focus on demonstrating the data-efficiency of our method.
> > >
> > > **Response to Q2:** Thank you for your suggestion. We follow your suggestion to conduct additional experiments with different temperatures and report both the average and best-of-N accuracies. The results are shown in the table below:
> > >
> > > The results show that temperature=0.5 yields the best performance for both methods. At temperature=2, neither method generates any fluent or valid sentences, resulting in an accuracy of 0. When the temperature is not overly deterministic (e.g., 0.5 and 1.0), FoR consistently outperforms SFT on both metrics, indicating that simply increasing the temperature for greater diversity is insufficient to match the accuracy improvements achieved by FoR. At temperatures=0, no diversity is permitted. In this case, SFT outperforms FoR to some extent, which is not unexpected as FoR is designed for diversity-solution settings.
> > >
> > >
> > > | Temperature  | 0     |               | 0.5   |               | 1.0   |               | 2.0   |               |
> > > |--------------|-------|---------------|-------|---------------|-------|---------------|-------|---------------|
> > > | **Method**   | avg Acc. (%) | best-of-N Acc. (%) | avg Acc. (%) | best-of-N Acc. (%) | avg Acc. (%) | best-of-N Acc. (%) | avg Acc. (%) | best-of-N Acc. (%) |
> > > | **SFT**      | 48.90       | 48.90         | 33.58       | 59.06         | 20.91       | 52.69         | 0.00        | 0.00          |
> > > | **FoR**      | 44.50       | 44.50         | 39.31       | 60.80         | 23.85       | 57.39         | 0.00        | 0.00          |

---

> > > > ### Comment · Reviewer_3ETe · 2024-11-27
> > > >
> > > > Thanks to the authors to providing the new results. I believe the proper evaluation is to compare performance using the best temperature for each method.
> > > >
> > > > For pass@1, taking the best temperature for both methods (0, 0), it appears that FoR is 4.4% worse than SFT.
> > > >
> > > > For pass@N, taking the best temperature for both methods (0.5, 0.5), it appears that FoR is 1.74% better than SFT.
> > > >
> > > > This does not seems like a significant improvement to me, so for this reason, I will keep my score the same.
> > > >
> > > > For future versions, I believe this work can be improved with the following:
> > > > 1) Focusing on tasks with more open ended responses in natural language.
> > > > 2) Conducting more thorough evaluations, e.g. comparing methods across different temperature settings for both pass@1 and pass@N.
> > > > 3) If the proposed method only show improvements in the low-data regime, then I think the paper should be MUCH more upfront about this. Otherwise, it would be great to also evaluate on the standard setting (training on the entire training set provided by a benchmark).
> > > > 4) Showing more significant margins of improvement.

---

> > > > > ### Author Response · Authors · 2024-11-30
> > > > >
> > > > > ### **New results showing large improvement margins**
> > > > >
> > > > > We have more time to run more comprehensive experiments on GSM8K and present the new results in the table below. Specifically, we report **accuracy of Pass@N** for N=1, 5, 10, and 20, at temperature=0.5 (as this temperature yielded the best performance for both methods in prior analysis). To ensure reliable results, all results are averaged over 5 random seeds, with the means and standard deviations provided. In addition, we also evaluate **diversity**, by following the same annotation rule as in our previous response (i.e., we manually annotated 50 questions, each with 20 samples generated by each method, and calculated the diversity metric as defined in Eq.12 of the paper).
> > > > >
> > > > > From the updated results, we can observe that our method outperforms SFT in terms of Pass@N across all N,  by **a notable margin of 3.5% - 6.6%**. For example, FoR achieves a **6.67%** absolute improvement over SFT in Pass@1 and a **3.74%** absolute improvement in Pass@20. The gains in Pass@5, Pass@10, and Pass@20, together with the improved diversity metric, underscore that our method can generate more diverse reasoning solutions and are more likely to succeed. We’ll add a more comprehensive comparison in the revised version.
> > > > >
> > > > >
> > > > > | Metric               | Pass@1 Acc. (%)   | Pass@5 Acc. (%)   | Pass@10 Acc. (%)  | Pass@20 Acc. (%)  | Diversity@20 |
> > > > > |----------------------|-------------------|-------------------|-------------------|-------------------|--------------|
> > > > > | SFT (Mean ± Std)     | 35.63 (2.03)      | 60.12 (1.20)      | 69.42 (0.41)      | 83.65 (0.48)      | 1.74         |
> > > > > | FoR (Ours, Mean ± Std) | 42.30 (4.52)     | 63.92 (3.17)      | 74.17 (1.65)      | 87.39 (0.87)      | 1.93         |
> > > > >
> > > > >
> > > > > ### **Clarification on temperature=0 results**
> > > > >
> > > > > In our last response, SFT performs better than FoR at temperature=0. However, we would like to emphasize that the focus of our work is on generating diverse reasoning solutions. A temperature=0 (i.e., greedy decoding) completely disables any diverse solutions, and thus is not an appropriate setting for our work.
> > > > >
> > > > > ### **Low-data regime**
> > > > >
> > > > > We have indeed mentioned the low-data regime setting up-front in the paper, such as in Abstract (Line.25) “*Extensive experiments show that, with limited training examples (e.g., 15 examples), …*”, in Introduction (Line.91) “*effective finetuning of LLMs to align with the task reward using only 15 input examples*”, and other places. Additionally, as described in Line.51, we have pointed out that one of the key advantages of our approach over SFT is that SFT “*often demands extensive supervision data to capture the full diversity of solutions*”, while our approach needs only limited data to produce diverse high-quality reasoning solutions. We will make this clearer in the revised version.

---

> ### Author Response · Authors · 2024-12-02
> **Gentle Reminder**
>
> Dear Reviewer,
> This is a gentle reminder that the discussion period will be closing soon. We have addressed all concerns by providing detailed clarifications and updated experimental results. Please don't hesitate to reach out if you have any additional questions. We would greatly appreciate it if you could consider raising the scores and/or engaging in further discussions. Thank you!

---

> > ### Comment · Reviewer_3ETe · 2024-12-02
> >
> > In your previous set of results, you showed Pass@1 accuracy at temperature 0.5 for SFT and FoR to be 33.58% and 39.31%, but your new results showed them to be 35.63% and 42.30%. I am confused about what changed?
> >
> > Your previous results also showed the Pass@1 accuracy at temperature 0 for SFT to be 48.90%, which is still better than the Pass@1 accuracy for FoR that you present in your new results (42.30%)?
> >
> > I understand that FoR is not designed for low temperatures. It is fine to not use low temperatures when presenting results for FoR. However, I believe **it is very important to use the best hyperparameters for baselines when conducting comparisons**. In this case, it means low temperature settings for SFT.
> >
> > Regarding the low data regime, the current writing makes it sound like it is a feature of the proposed method that it works with with limited data. However, most realistic finetuning tasks have more than 15 examples. If the proposed method does not work better than baselines when there are more finetuning examples (e.g. thousands), then this should be more clearly stated as a **limitation**, and the problem setting should be scoped to be much more limited to settings where finetuning examples are extremely rare.

---

> ### Author Response · Authors · 2024-12-03
>
> Thank you so much for your feedback!
>
> &nbsp;
>
> "**use the best hyperparameters for baselines when conducting comparisons**"
>
> We agree with your comment that during comparison, the best hyperparameters for baselines should be used. **This is indeed what we've done in the paper.** For example, in Table.1, we've compared with SFT under different temperatures (0.1, 0.5, 1.0). We've also tried different decoding algorithms (e.g., DBS, Nucleus, Typical) to see which setting is the best for the SFT baseline. Results show that our method outperforms SFT with the best hyperparameter configuration.
>
> On GSM8K above, SFT (temperature=0) achieves better accuracy than our method, but SFT (temperature=0) can produce only one solution at maximum, while ours aims at generating more diverse solutions. We see **diversity** as a major advantage of our approach.
>
> &nbsp;
>
> **New results**
>
> As stated in the last response, the new results were obtained by averaging over 5 runs with different random seeds. So the numbers could be slightly different from (and are more robust than) previous results from only one run.
>
> &nbsp;
>
> **Low-data regime**
>
> Thanks for the comment! Low-data setting is an important scenario in practice. In addition to **finetuning-based** methods, we've also compared with **prompting-based** methods that are commonly used in low-data settings (e.g., see Table.1). Results show our method outperforms both finetuning- and prompting-based methods. We thus see our ability of learning to produce diverse solutions from only limited examples as an advantage. We'll add more discussion and results especially in the settings when more (e.g., thousands of) data are available. Thanks again for the suggestion.

---

### Official Review · Reviewer_waXE · 2024-11-04

**Soundness:** 4
**Presentation:** 1
**Contribution:** 3
**Rating:** 8
**Confidence:** 3

**Summary:**

The paper presents Flow of Reasoning (FoR), a novel fine-tuning approach for LLMs that aims to enhance solution diversity. FoR frames the reasoning process as a Markovian flow, sampling diverse reasoning paths with probabilities aligned to the reward associated with each target problem similar to GFlowNet. Extensive experiments across multiple domains and baselines demonstrate that the proposed approach helps LLMs discover more diverse and higher-quality answers, showcasing the effectiveness of the algorithm.

**Strengths:**

1. The paper addresses the important challenge of encouraging diverse reasoning paths, a key problem in advancing LLMs’ capability for creative and robust problem-solving.
2. The experimental setup is solid, with evaluations across a wide range of domains and baseline models. Additionally, the comprehensive ablation studies effectively demonstrate the effectiveness of the design choices.
3. Strong performance is demonstrated, with the proposed approach outperforming state-of-the-art methods, including both prompt-based and fine-tuned models. In some tasks, the method even achieves results comparable to the advanced o1 reasoning model, despite using a significantly smaller Llama 3 model.

**Weaknesses:**

### The Presentation of the Paper Can Be Improved

- The motivation of the method is not clear enough. While the authors emphasize formulating the problem as a Markovian flow and sampling reasoning paths proportionally to rewards by drawing from GFlowNets, it is not clear why these specific elements are essential or expected to be effective in addressing the problem of diverse reasoning paths.
- The authors could consider presenting the motivation and broader rationale behind key design choices before delving into technical and mathematical specifics. For instance, Section 3.2 is challenging to follow without a detailed explanation of why and how choices like the trajectory balance approach, log-variance approximation, or local search are adopted. It would also be helpful to clarify the reasoning behind Equations (5) and (7) and the role of local search within the approach.
- While the empirical results are impressive, the paper would benefit from additional analysis to strengthen its scientific arguments. For example, a case study illustrating the diverse reasoning paths generated by FoR, which other methods fail to find, would be informative. Additionally, a discussion on why FoR achieves such high performance while focusing on diversity would be valuable, especially considering that in most experiments, it outperforms baselines that emphasize mostly reasoning path quality. It would also be interesting to analyze how diverse reasoning paths may contribute to FoR’s effectiveness in out-of-distribution tasks.

### Limitations of the Markov Assumption and Markovian Flow Setup

The core setup of the algorithm relies on Markov assumptions and a Markovian flow structure, which are suitable only for domains that can accommodate state/action flows under these constraints. However, many complex real-world reasoning tasks do not necessarily align with the Markov assumption, having significant partial observability challenges or making it challenging to define clear “state” and “action” flows. More discussions of these limitations would help readers better understand the specific types of domains where the proposed method is most applicable and where it may face constraints.

### Challenges of Hard Exploration and Sparse Rewards

The method’s core approach—sampling reasoning paths proportionally to rewards—faces inherent challenges in many real-world decision-making problems, where exploration is difficult, and rewards are sparse. In scenarios where only a binary success/failure reward is available, for example, the proposed method may converge toward behavior similar to traditional reinforcement learning, as it becomes difficult to maintain diverse reasoning paths with limited reward signals.

To clarify, I recognize that it is unrealistic to expect a single paper to address all the challenges present in the field, and this paper has already made valuable contributions to a specific class of problems. However, discussing these broader challenges would benefit readers by clarifying how this method could perform in more complex, real-world applications. Could the current approach address such challenges, or might these issues require further research and adaptation of the proposed method? Alternatively, is this method perhaps less suitable for tasks with particularly sparse rewards?

### See other questions below. Some of those questions may be worth discussing.

**Questions:**

1. Why does only the Game of 24 domain use a different model?
1. Could the authors clarify which components are borrowed from existing works and which are newly proposed in this method? Many core components (sampling proportional to rewards, trajectory balance objective, adaptive efficient exploration) seem to be derived from existing approaches.
1. Could the authors clarify the usage of examples in the test dataset? Is it used for few-shot prompting during inference, or is it only used for calculating creativity metrics?
1. In Line 301, what is the value of N for each domain? The choice of N seems to not be specified for domains like Blocks World or Game of 24.
1. While it's clear that each fine-tuning method shares the same offline dataset, has the size of the online dataset also been controlled across methods?
1. The proposed method seems to not include search during inference. Why then is its runtime significantly higher than other COT/SFT methods that also do not include search?
1. For baseline evaluation, do the prompts explicitly include keywords to encourage diverse responses? If not, this could be worth investigating.

---

> ### Author Response · Authors · 2024-11-21
> **Response to Reviewer waXE -  Part 1**
>
> Thank you very much for your insightful comments and suggestions! Below, we address each of the comments in detail.
>
> ## Response to Weaknesses:
>
> >W1.1: The motivation behind the method lacks clarity. While the authors focus on framing the problem as a Markovian flow and sampling reasoning paths using GFlowNets, it is unclear why these specific elements are crucial or effective for addressing the issue of diverse reasoning paths.
>
> **Response-W1.1:**
>
> The cited prior work referenced in Line.138 [1] demonstrates that GFlowNets are specifically designed to generate *diverse and high-quality* samples with probabilities proportional to their rewards. This mechanism encourages exploration because it allows the model to assign probabilities to all trajectories based on their “goodness” (reward), rather than focusing solely on a single optimal path. Therefore, GFlowNets can explore a broader solution space, discovering multiple ways to complete a task during sampling. This contrasts with methods like SFT or PPO (Line.131), which tend to constrain the diversity of generated solutions due to reward-maximization. Building on this intuition, our work is the first to formulate multi-step LLM reasoning as Markovian flow, which makes it possible to adapt GFlowNets approaches into our new settings for diverse reasoning paths. We will revise this part to be more clear in the next version.
>
> ---
>
> >W1.2: The authors could consider presenting the motivation and broader rationale behind key design choices before delving into technical and mathematical specifics.
>
> **Response-W1.2:**
>
> Thanks for the suggestion! We would first like to note that one of contributions of our work lies in formulating LLM reasoning as a Markovian flow. This formulation enables us to connect to *existing successful* approaches (proposed in other settings) and adapt them to the new multi-step LLM reasoning setting. We would like to highlight that the ability to re-use and generalize *existing successful* approaches to our new problem setting (instead of having to re-invent all details from scratch) is a key advantage of our work.
>
> Below we clarify motivations of each component, and will make this clearer in the revised paper.
> 1. Why trajectory balance?
>
> As noted in Line.233, the trajectory balance objective has demonstrated greater efficiency than other learning objectives, like Detailed Balance (DB) mentioned in Appendix B, Line.207. We’ll add more details in the next version.
>
> 2. Why log-variance approximation?
>
> Estimating $Z$ in the trajectory balance loss requires an additional model beyond the LLM policy, which is computationally cumbersome. As noted in Line.246, log-variance approximation provides a practical and effective alternative.
>
> 3. Why local search?
>
> The vast trajectory space and sparse rewards make generating high-reward samples consistently challenging. Local search, as mentioned in Line.262, helps explore high-reward regions in the sample space. The ablation studies in Section 4.7 further validate the effectiveness of this approach.
>
> ---
>
> >W1.3.1: Benefit from a case study illustrating the diverse reasoning paths generated by FoR, which other methods fail to find, would be informative.
>
> **Response-W1.3.1:**
>
> Please refer to the **Qualitative Results** part in the **General Response**.
>
> ---
>
> >W1.3.2: a discussion on why FoR achieves such high performance while focusing on diversity would be valuable.
>
>
> **Response-W1.3.2:**
>
> FoR achieves high performance through its inherently **exploratory nature** [2], which uncovers multiple ways to accomplish a task, with each solution being sampled in proportion to its "goodness" or reward. This ensures diverse, effective solutions are consistently represented. As shown in the **Qualitative Results** part in the **General Response**, SFT samples often get stuck on specific incorrect trajectories, repeatedly failing to explore alternative solutions, whereas our method effectively discovers multiple valid trajectories, ensuring diverse exploration and avoiding over-reliance on a single trajectory.

---

> ### Author Response · Authors · 2024-11-21
> **Response to Reviewer waXE - Part 2**
>
> >W1.3.3 It would also be interesting to analyze how diverse reasoning paths may contribute to FoR’s effectiveness in out-of-distribution tasks.
>
> **Response-W1.3.3:**
>
> Thanks for the insightful suggestion! By exploring multiple high-reward trajectories and matching the reward distribution, FoR develops a richer understanding of the underlying task structure, rather than overfitting to specific patterns seen during training (e.g., maximizing reward as in RL). This diversity facilitates the model to adapt to novel scenarios on OOD problems, as it can leverage alternative reasoning paths that may still align with the new data distributions. [2,7]
>
> To further assess performance on out-of-distribution (OOD) settings, we conduct experiments on the BlocksWorld task (Section.4.2). Specifically, we train the model using FoR and SFT on a 2-step training set and evaluate them on a 4-step test set, and train the model on 4-step training set and evaluate them on the 6-step test set. This allows us to analyze their generalization on OOD problems. For prompting-based baselines, we use 2-step and 4-step examples as demonstrations, respectively.
>
> According to the table below, FoR maintains highest accuracy (71.43%) on OOD tasks compared to other methods like CoT and SFT, which range from 9.52% to 14.28%. FoR also achieves greater diversity (by an absolute improvement of 0.2 over SFT), highlighting its superior generalization and solution exploration capabilities.
>
>
> |               |        |         2-step to 4-step                 |                           |       |                  4-step to 6-step         |                           |
> |---------------|-------------------------|--------------------------|---------------------------|-------------------------|--------------------------|---------------------------|
> | Method        | Acc. (%)                | Diversity                      | Creativity (%)            | Acc. (%)                | Diversity                      | Creativity (%)            |
> | CoT (1-shot)  | 9.52          | 1.0                      | 3.12                      | 2.02                   | 1.0                      | 0                         |
> | CoT (5-shot)  | 14.28          | 1.0                      | 3.12                      | 12.12                  | 1.08                     | 3.45                      |
> | CoT (15-shot) | 11.90          | 1.0                      | 3.12                      | 8.08                   | 1.0                      | 0                         |
> | ToT (BFS)     | 9.52            | -                        | -                         | 8.08                   | -                        | -                         |
> | ToT (DFS)     | 4.76           | -                        | -                         | 6.06                   | -                        | -                         |
> | RAP           | **80.95**         | -                        | -                         | 34.34                  | -                        | -                         |
> | SFT (a=1.0)   | 11.92          | 1.0               | 9.37                      | 28.28                  | 1.03                     | 1.15                      |
> | FoR (Ours)           | 71.43          | **1.20**             | **59.38**                     | **65.65**                  | **1.25**                     | **60.92**                     |
>
> ---
>
> >W2: The core setup of the algorithm relies on Markov assumptions and a Markovian flow structure, which are suitable only for domains that can accommodate state/action flows under these constraints. However, many complex real-world reasoning tasks do not necessarily align with the Markov assumption, having significant partial observability challenges or making it challenging to define clear “state” and “action” flows.
>
> **Response-W2:**
>
> The Markovian assumption in our approach, that the next state is only based on the previous state, is indeed inspired by the common Reinforcement Learning (RL) formulations. The same state-action formulation in RL has been successfully applied to various LLM reasoning tasks [3] (e.g., with proper definition of “state” to include necessary reasoning history). Consequently, our formulation inherits this compatibility, allowing it to be applied across a wide range of LLM reasoning scenarios.
>
> While being as general as RL approaches, we highlight that the key advantage of our approach over traditional RL formulations is its diversity-seeking nature by encouraging exploration and sampling of diverse reasoning paths.

---

> ### Author Response · Authors · 2024-11-21
> **Response to Reviewer waXE - Part 3**
>
> >W3: The method’s core approach—sampling reasoning paths proportionally to rewards—faces inherent challenges in many real-world decision-making problems, where exploration is difficult, and rewards are sparse. In scenarios where only a binary success/failure reward is available, for example, the proposed method may converge toward behavior similar to traditional reinforcement learning.
>
> **Response-W3:**
>
> For tasks where ground-truth dense rewards are unavailable (such as Blockworlds and Game24 in our experiments), we acknowledge that this presents difficulties for traditional GFlowNets and also existing RL algorithms like PPO. To address this, we employ *augmented rewards* (see "Reward Design" sections in every task) which help make the reward signal more dense, and incorporate the technique of *local search* to expand the exploration of trajectories during online training (Section 3.2.2 and Appendix D and E). Our ablation study (Section 4.7) demonstrates that these techniques effectively enhance exploration during training, mitigate the challenges of sparse rewards, and indeed lead to greatly improved performance and diversity than traditional RL (e.g., Table 1).
>
> We acknowledge that for tasks or areas beyond the internal knowledge of LLMs, the augmented reward signal may be less effective—a challenge that similarly affects RL methods. Future work could explore incorporating measures of uncertainty or reliability in LLM knowledge into reward design. For example, uncertainty-aware rewards [4] could adaptively guide exploration in unfamiliar areas, enabling the model to better handle tasks with sparse or unreliable reward signals.
>
>
>
> ---
>
>
>
> ## Response to Questions
>
> >Q1: Why does only the Game of 24 domain use a different model?
>
> **Response-Q1:**
>
> We use LLaMA-2-13B for Game of 24 as it achieves reasonably good performance across all comparison methods. We are currently running additional experiments using LLaMA3-8B (the same model as in other tasks) on the Game of 24 and will include these results in the revised paper.
>
>
> ---
>
>
> >Q2: Could the authors clarify which components are borrowed from existing works and which are newly proposed in this method? Many core components (sampling proportional to rewards, trajectory balance objective, adaptive efficient exploration) seem to be derived from existing approaches.
>
> **Response-Q2:**
>
> Please refer to the **Contribution** part in the **General Response**. Our method models the multi-step reasoning problem as a Markovian flow. This new formulation of multi-step LLM reasoning allows us to adapt and integrate existing successful approaches. As mentioned above, we believe the ability to incorporate existing techniques is an advantage of our formulation.
>
> ---
>
> >Q3: Could the authors clarify the usage of examples in the test dataset? Is it used for few-shot prompting during inference, or is it only used for calculating creativity metrics?
>
>
> **Response-Q3:**
>
> We didn’t use the test data as part of the few-shot prompting prompt. And yes, the test set is only used for calculating the creativity scores.
>
> ---
>
> >Q4: In Line 301, what is the value of N for each domain? The choice of N seems to not be specified for domains like Blocks World or Game of 24.
>
>
> **Response-Q4:**
>
> We would like to clarify that the value $n$ refers to the number of samples generated during inference, as described in the **Setup** section (Section 4) of each task. For example, $n$ is 8, 20, and 40 for the 2, 4, and 6-step cases, respectively, for BlocksWorld (see Line.323) and $n$ is 20 (see Line.373) for Game of 24.
>
> ---
>
> >Q5: While it's clear that each fine-tuning method shares the same offline dataset, has the size of the online dataset also been controlled across methods?
>
> **Response-Q5:**
>
> Yes,  we sample the same amount of online data for training FoR and the baselines like SFT+PPO and SFT + GFN-CoT, we will make it more clear in the experimental setting in the revision.

---

> ### Author Response · Authors · 2024-11-21
> **Response to Reviewer waXE - Part 4**
>
> >Q6: The proposed method seems to not include search during inference. Why then is its runtime significantly higher than other COT/SFT methods that also do not include search?
>
> **Response-Q6:**
>
> Following LLM-Reasoners [5], our method (and the baselines like RAP [6]) utilize another LLM (e.g., in BlocksWorld) or additional environment (e.g., in Rubik's Cube) to perform the state transition $T$ (Line.218-220) while the CoT/SFT method only predicts the action series. This results in different inference times for these methods.
>
> ---
>
> >Q7: For baseline evaluation, do the prompts explicitly include keywords to encourage diverse responses? If not, this could be worth investigating.
>
> **Response-Q7:**
>
> We did not include these keywords in our prompt to reflect their common usage in practice and to ensure a fair comparison across all baselines. We follow your suggestion and add a diversity-encouraging prompt as instruction:
>
> - “Please carefully understand the goals and initial states, then come up with diverse solutions and think outside the box.”
>
> We evaluate multiple baseline methods using LLama-3-8B as the base model, following the exact same settings described in our paper. The results for BlocksWorld are reported below. The numbers in parentheses indicate the performance difference compared to the original prompt without the diversity-encouraging instruction.
>
>
> We observe that diversity-encouraging prompts for the CoT and SFT baselines lead to improvements in both diversity and accuracy, with average absolute gains of 0.03 and 5.11%, respectively. However, FoR still outperforms them, achieving average absolute improvements of 0.19 in diversity, 9.46% in creativity, and 34.93% in accuracy compared to the best baseline for each metric. These results will be included in our next version.
>
> | Method           |                   |           4-step           |                      |                   |           6-step           |                      |
> |------------------|----------------------------|----------------------|----------------------|----------------------------|----------------------|----------------------|
> |                  | Acc. (%)           | Diversity    | Creativity (%)  | Acc. (%)           | Diversity    | Creativity (%)  |
> | CoT (1-shot)     | 16.67 (-10.90)                     | 1.00 (-0.05)               | 0.0   (0.00)               | 11.11     (-4.71)                 | 1.09   (+0.04)             | 0.0  (0.00)                |
> | CoT (5-shot)     | 59.52 (+16.66)                     | 1.12 (+0.08)               | 2.04  (+2.04)               | 33.33  (+3.70)                    | 1.03  (+0.00)              | 0.79   (+0.79)              |
> | CoT (15-shot)    | 52.38  (+12.32)                    | 1.09  (+0.06)              | 0.0    (0.00)              | 13.13 (-6.40)                     | 1.07  (+0.04)              | 0.0   (0.00)               |
> | SFT ($\alpha$=1.0)      | 59.52  (+17.46)                    | 1.10  (+0.05)              | 0.0    (0.00)              | 47.47   (+12.79)                   | 1.10  (+0.06)              | 0.0    (0.00)              |
> | FoR (Ours)             | **98.41**                      | **1.27**                | **12.24**                | **78.44**                      | **1.33**                | **9.52**                |
>
> ---
> References:
>
> [1] Bengio, Yoshua, et al. "Gflownet foundations." The Journal of Machine Learning Research 24.1 (2023).
>
> [2] Madan, Kanika, et al. "Goal2FlowNet: Learning Diverse Policy Covers using GFlowNets for Goal-Conditioned RL."
>
> [3] Wang, C., Deng, Y., Lyu, Z., Zeng, L., He, J., Yan, S., & An, B. (2024). Q*: Improving multi-step reasoning for llms with deliberative planning. arXiv preprint arXiv:2406.14283.
>
> [4] Lou, Xingzhou, et al. "Uncertainty-aware reward model: Teaching reward models to know what is unknown." arXiv preprint arXiv:2410.00847 (2024).
>
> [5] Hao, Shibo, et al. "LLM Reasoners: New Evaluation, Library, and Analysis of Step-by-Step Reasoning with Large Language Models." First Conference on Language Modeling.
>
> [6] Hao, Shibo, et al. "Reasoning with Language Model is Planning with World Model." Proceedings of the 2023 Conference on Empirical Methods in Natural Language Processing. 2023.
>
> [7] Hu, Edward J., et al. "Amortizing intractable inference in large language models." The Thirteenth International Conference on Learning Representations 12 (2024) Oral.

---

> ### Author Response · Authors · 2024-11-25
>
> Dear Reviewer,
>
> This is a gentle reminder that the discussion period is closing soon. We've responded to all concerns, providing detailed clarifications and new experimental results. Please feel free to let us know if you have any further questions. We'd greatly appreciate it if you could consider raising scores and/or engaging in further discussions. Thanks!

---

> ### Comment · Reviewer_waXE · 2024-11-25
>
> Thank you for your thorough response to my questions. I find the added experimental results insightful, and the responses comprehensively address my primary concerns. My sole remaining reservation is that the paper incorporates multiple components that might challenge readers' understanding of each component's contribution to the main scientific argument. Nonetheless, the authors have conducted extensive ablation studies that effectively demonstrate the significance of these components, substantiating their critical role in the proposed algorithm. Overall, the paper presents solid empirical evidence supporting the algorithm's effectiveness and makes a significant contribution to an important yet underexplored domain of LLM reasoning, i.e. the exploration of diverse reasoning paths in LLMs. I will recommend the paper for acceptance and adjust my score accordingly.

---

> > ### Author Response · Authors · 2024-11-26
> > **Thank you!**
> >
> > Thank you so much for the encouraging and positive feedback! We'll improve the paper further to make the components clearer. Thanks again!

---

### Author Response · Authors · 2024-11-21
**General Response - Part 2**

## Qualitative Results

Our paper already provides case study examples in both Figure 1 and Appendix F. In Figure 1’s (left) Game24 example, we show that the SFT and CoT methods can only find one trajectory, while our FoR can find 4 different trajectories. Appendix F provides more examples of FoR among different tasks.

We would like to present two additional qualitative results on the Game24 task to highlight the strengths of our method. This part would also be added to our revision.


- **Balance between diversity and accuracy**: For the first case study (please see the figure [here](https://anonymous.4open.science/r/iclr25-432C/casestudy.jpg)), we use the problem *(3,4,6,11)* to show how FoR achieves such high performance while focusing on diversity. As illustrated in the figure, we compare trajectories sampled 20 times by both SFT and FoR. While both methods produce diverse trajectories initially, FoR demonstrates better capability in reaching successful final steps from various middle steps. For example, FoR successfully transitions from intermediate steps such as *(3,6,15)* to the target 24, whereas SFT fails to do so. This highlights the effectiveness of FoR’s design in simultaneously promoting diversity and ensuring accuracy.

- **Better robustness due to exploratory nature**: For the second case study (please see the figure [here](https://anonymous.4open.science/r/iclr25-432C/casestudy2.jpg)), we use the problem *(7,9,9,13)* to demonstrate the robustness of FoR. As shown in the figure, SFT repeatedly fails by getting stuck in a single second state of *(9,3,16)* 20 times, while FoR successfully discovers multiple diverse trajectories leading to the correct solution. This robustness can be attributed to the exploratory nature of FoR's training objective, which encourages the model to sample diverse successful trajectories. By expanding the search space through high-reward exploration, FoR increases the chance to find successful outcomes. This capability not only improves the robustness of the model but also enhances its generalization to new scenarios, showcasing the effectiveness of FoR in addressing complex reasoning tasks.

---

References

[1] Hu, Edward J., et al. "Amortizing intractable inference in large language models." The Thirteenth International Conference on Learning Representations 12 (2024) Oral.

[2] Yao, Shunyu, et al. "Tree of thoughts: Deliberate problem solving with large language models." Advances in Neural Information Processing Systems 36 (2024).

[3] Shibo Hao, et al. Reasoning with Language Model is Planning with World Model. In Proceedings of the 2023 Conference on Empirical Methods in Natural Language Processing, pages 8154–8173, Singapore. Association for Computational Linguistics.

---

> ### Comment · Reviewer_h3sb · 2024-11-28
> **Concerns about the 'Contributions' and 'Qualitative Results'**
>
> You refer to the contribution and the qualitative results as the answer to the first weakness proposed by the reviewer h3sb. However, after carefully reviewing the authors' responses, I believe that the current answers may be insufficient to clearly convey the major insight or observation derived from this submission. When discussing insights, it may not be merely about integer numbers or demonstrations but rather the ideas and conclusions that this paper offers and how they contribute to the community. For example, when focusing on the diversity of LLMs, what is the impact of this new objective on accuracy, and to what extent? Is there a tradeoff between the two, or does improving diversity lead to higher accuracy? What are the underlying reasons or observations behind these outcomes? What are the in-depth insights into the corresponding designs?
>
> For the final question, what confused me was that, upon reading the paper, many concepts and observations discussed appeared to originate from existing works, leading the authors to cite numerous papers in each part of the design section. Additionally, the authors made only minor direct modifications. As a result, as I previously mentioned, these limitations make the paper resemble more of a technical implementation rather than an academic contribution.
>
> The numbers presented by the authors in the paper and the rebuttal are certainly an important aspect, but it is even more important to present new findings and observations that can inspire other researchers in the field.
>
> With more insights and discussions presented in the paper, the numbers would become more convincing.
>
> I believe that the lack of in-depth insights and well-organized motivations in the method design sections is likely why many reviewers assigned a negative score to the 'presentation' category. Reading the paper feels more like reading a technical report on a project.

---

> > ### Author Response · Authors · 2024-11-29
> > **Response to Reviewer h3sb**
> >
> > Thanks for the comments.
> >
> > In the responses to your review (h3sb), we not only referred to the contributions and qualitative results, but also articulated the motivations and intuitions of the method design, as well as on why this design works to improve both reasoning accuracy and diversity. For example, in response to your question, “Why sampling in such a way can generate not only diverse but also high-quality reasoning paths?” We explained that encouraging the model to explore more potential reasoning trajectories (guided by reward) makes it more likely to find correct solutions instead of getting stuck in a specific incorrect step—a common problem in multi-step LLM reasoning. We then provided the qualitative demonstration with a real running example to further illustrate this insight (e.g., how previous methods like SFT got stuck).  We welcome more specific questions regarding intuition, mathematical formulation, and/or results.
> >
> > We respectfully disagree with the assertion that we “made only minor direct modifications”. We’ve clarified technical misunderstanding from you (e.g., clarifying MCTS is indeed different from Markovian Flow), and articulated our specific contributions.
> >
> > We’d like to point out that other reviewers have appreciated the presentation:
> > - Reviewers 3ETe and iYFq both gave a presentation score of “3 good”, and commented “Paper is organized cleanly and easy to understanding” and “well-written and easy to follow”, respectively.
> > - Reviewer waXE gave concrete suggestions on the presentation, and after our responses, has raised their overall rating to “8: accept, good paper”.

---

> > > ### Comment · Reviewer_h3sb · 2024-12-02
> > > **Further Comments**
> > >
> > > Thank you very much for the further discussion.
> > >
> > > Again, I would like to emphasize that you make a very strong assumption and assertion that your work is the first and only exploration of this problem, but you lack a specific and unique definition to substantiate this claim. Therefore, I use the cautious and argumentative tone of "may" to highlight that such repeated emphasis introduces the risk of over-claiming in the paper. For the four papers I mentioned, I would like to point out that, due to the inherent multi-step reasoning ability of LLMs, which has been established as a core capability, diversity has been explored in existing research either to enhance ensemble approaches or to directly improve accuracy. Besides, some papers explicitly highlight the goal of enhancing diversity. For instance, in [2], they discussed the relationship between the performance and the diversity of negative reasoning processes in LLMs. When LLMs inherently perform multi-step reasoning, exploring such diversity contributes to problem-solving by improving both diversity and accuracy.
> > >
> > > Therefore, I would like to remind the author of this submission to provide a more direct and clearer explanation in the paper regarding what specifically is being claimed as the first and why. In the current submission, I find it difficult to discern the progress made in addressing diversity in the application and reasoning of LLMs. When diversity is positioned as the core motivation of this paper, it is unclear why the related work section does not include a specific subsection to discuss and compare existing research on diversity.
> > >
> > > By saying "they all formulate multi-step LLM reasoning from a Markovian flow perspective.", I was not to emphasize the equivalence of the two. Instead, I want to mention that these methods indeed conceptualize the reasoning process as a sequence of states where the next state depends only on the current state, adhering to the Markov property. Therefore, I do not observe the major insights of why using Markovian flow is the core contribution. Is this only because you apply the existing framework proposed in GFlowNet Foundations to the multi-step reasoning of LLMs? In summary, while presenting these existing MCTS-based approaches, I aim to highlight that stating, 'We are the first to formulate multi-step LLM reasoning from a Markovian flow perspective,' may be challenging to substantiate. This is because much of your work appears to apply existing frameworks rather than introducing significant novel contributions. I cannot fully capture what specifically makes this approach novel.
> > >
> > > Furthermore, by stating that the presentation falls below the borderline for an ICLR conference, I aim to emphasize not only the points raised in review (waXE) but also those highlighted in my own review block (h3sb). The motivation of this submission is not well-organized. The insights of this paper should be improved. The motivation for this submission is not well-structured, and the insights provided in the paper need further development. For instance, what specific factors enable this submission to achieve the best performance in both accuracy and diversity, especially when other approaches primarily focus on improving accuracy while you place greater emphasis on diversity?
> > >
> > > From my perspective, this paper appears to directly apply existing work, such as the approach outlined in GFlowNet Foundations, to the multi-step reasoning of LLMs. Surprisingly, this results in state-of-the-art performance in both accuracy and diversity. Moreover, most of the equations and notations in this submission are directly derived from existing methods. Similarly, many of the objectives and mathematical explanations are also borrowed from previously established approaches. This is what I mean by saying the presentation of this submission should be improved significantly.
> > >
> > > In addition, every time I revisit subsection 3.1 of the paper, I find it completely unclear what contributions are original to the authors and what is derived from existing works. Does the content primarily aim to demonstrate how the authors 'seamlessly integrate existing successful GFlowNet training methods? This is what I mean when I say that __this work feels more like a technical report on applying an existing framework to the multi-step reasoning of LLMs__.
> > >
> > > In summary, I __hold my view__ that this work, which claims too many firsts and breakthroughs but lacks the necessary definitions and discussions of related work, uses too much content from existing work (formulas, definitions, mathematical explanations), and directly presents the best numerical results but lacks convincing discussions and insights, __does not reach the borderline ICLR__. Therefore, I believe that 'marginally below the acceptance threshold' is a fair assessment, particularly after thoroughly reviewing this submission multiple times.

---

### Author Response · Authors · 2024-11-21
**General Response - Part 1**

We sincerely thank all reviewers for their encouraging feedback on our paper. We are grateful for the recognition of our contributions, including our novel focus on the **diversity of multi-step LLM reasoning**, an area previously unexplored, and the effective solution to this challenge (*h3sb*); our innovative idea of modeling multi-step reasoning as Markovian flow and applying GFlowNets to the LLM reasoning tasks (*iYFq*), which leads to significant improvements of success rate from 20% to 80% for a 7B model, even surpassing OpenAI-o1 model in some tasks (*h3sb*, *3ETe*, *waXE*). We are pleased that the reviewers recognize the robustness of our solid experiments across 5 benchmarks and ablation studies (*waXE*, *3ETe*, *h3sb*), as well as the well-organized and easy-to-follow presentation of our paper (*3ETe*, *iYFq*). We also appreciate the acknowledgment of the data efficiency demonstrated by our method (*h3sb*).



## Contributions

We would like to highlight again the **contributions** of our work that differ from previous works as follows:

1. **Novel problem:** We are the first to introduce the problem of **diverse** LLM reasoning. This problem is fundamental to building more intelligent machines and is crucial for applications such as creative problem-solving, robustness, and scientific discovery.

2. **Technical contributions (I):** We are the first to formulate multi-step LLM reasoning from a Markovian flow perspective. Unlike the latest work (such as [1]) which focuses on token-level modeling (Line.92), our state-action formulation is more general and better suited for complex multi-step reasoning settings. On The Other Hand, compared to recent Reinforcement Learning-based LLM reasoning (which also adopts state-action formulation and thus is similarly general) [2,3], our Markovian flow perspective allows for promoting diverse reasoning solutions, rather than solely optimizing for the maximum-reward solution.

3. **Technical contributions (II)**: Based on the above Markovian flow formulation, we manage to adapt and integrate a series of the latest GFlowNets approaches to our problem setting, such as *trajectory balance* objective with *log-variance* approximation for tractability, and design new *local search* method for better solution exploration.

4. **Empirical contributions:** We conduct extensive experiments across **5 benchmarks** challenging for LLMs and approximately **20 baseline methods**, showing our approach achieves substantial improvements not only in accuracy scores (by a 20% to 80% margin) but also in diversity and creativity. Note that *we are the first to systematically measure diversity and creativity in multi-step LLM reasoning*. In the task of Blocksworld, our approach with a 7B model either approaches or exceeds the performance of OpenAI-o1.

---

> ### Comment · Reviewer_h3sb · 2024-11-27
> **Concerns About the 'Novel Problem' and 'Contributions'**
>
> Thank you very much for adding additional explanation. Thanks to the extension of the ICLR review process, I now have more time to thoroughly read the paper and the rebuttal again.
>
> I am concerned about the "Novel Problem," as it is directly related to my Weakness1.1. More concerns will be outlined in the review block unless the authors address my other questions directly here.
>
> It may introduce the risk of __over-claiming__ by directly emphasizing and highlighting decisive conclusions such as "_an area previously unexplored_" and "_We are the first to introduce the problem_".
>
> It may be better to provide a very narrow and precise definition of diversity; otherwise, repeatedly emphasizing that it is the first in the community may be perceived as unfair.
>
> For example, considering the diversity explored in existing works [1][2][3][4], these studies have all contributed to the community by highlighting the importance and application of diversity in LLMs to some extent.
>
> Therefore, I encourage the author to provide a stricter and more narrowly defined explanation of diversity and to exercise greater caution when making conclusions such as "unexplored" and "first."
>
> It may even lead to greater misunderstanding and confusion when the authors emphasize, "_We are the first to formulate multi-step LLM reasoning from a Markovian flow perspective_."  The main reason is that Markovian flow is not a novel idea but a well-established domain that has been explored for many years. Moreover, in the context of LLMs, numerous works have applied Monte Carlo Tree Search (MCTS) to formulate or organize the step-by-step reasoning process of LLMs. We can say that "they all formulate multi-step LLM reasoning from a Markovian flow perspective."
>
> Therefore, once again, it may be unfair to directly conclude that "we are the first" without a very clear and narrowly defined declaration. Personally, I believe that the risk of over-claiming may significantly reduce the persuasiveness of the paper.
>
>
> [1]. DLCRec: A Novel Approach for Managing Diversity in LLM-Based Recommender Systems, WSDM 2025
>
> [2]. LLM-TOPLA: Efficient LLM Ensemble by Maximising Diversity, EMNLP 2024.
>
> [3]. Exploring the Capabilities of Large Language Models for Generating Diverse Design Solutions, 2024.
>
> [4]. Diversity of Thought Improves Reasoning Abilities of LLMs, 2023.

---

> > ### Author Response · Authors · 2024-11-29
> > **Response to Reviewer h3sb**
> >
> > Thank you for your time and engaging in our discussion.
> >
> > **Novel Problem**
> >
> > We’d like to clarify that our focus is on diversity specifically in the context of LLM multi-step reasoning, where the LLM generates reasoning trajectories step-by-step until reaching final answers, as shown in Figure.1 We’ve highlighted this specific setting in many places throughout the paper, such as Abstract (Line.15 “existing approaches to multi-step reasoning with LLMs”; Line.21 “FoR formulates multi-step LLM reasoning as …”), Figure.1 (Line.70 “Figure 1: Multi-step LLM reasoning as …”), Introduction (Line.83 “FoR enables diversity-seeking
> > finetuning of multi-step LLM reasoning ...”), etc.
> >
> > Thanks for pointing to the existing works. They have different focuses and solve different problems than ours:
> >
> > - [1] studies diversity in recommendation items, which is a different problem than our diverse multi-step LLM reasoning.
> > - [2] focuses on improving ensemble accuracy by leveraging error diversity across multiple LLMs. In contrast, we aim to generate diverse reasoning trajectories with a single LLM, focusing on not only accuracy but also diversity and creativity.
> > - [3] focuses on generating different design solutions in human-computer interaction (HCI) settings, which is different from multi-step LLM reasoning.
> > - [4], like [2], focuses only on improving LLM accuracy using diverse prompts. Our work aims to improve diversity and creativity, with explicit quantitative evaluation on these aspects. Notably, [reviewers of [4]](https://openreview.net/forum?id=FvfhHucpLd&referrer=%5Bprofile%20of%20Ranjita%20Naik%5D(%2Fprofile%3Fid%3D~Ranjita_Naik1)) have exactly questioned this work for its lack of evaluation/analysis of generation diversity.
> >
> > In sum, these works are related to LLM generation diversity in general, but are very different from our specific focus in the setting of LLM multi-step reasoning. We’ve never claimed we’re the first on LLM diversity in general. Indeed, our paper has cited/discussed many existing diverse LLM generation works, such as (Hu et al., 2023a; Malkin et al., 2022a), as well as empirically compared with extensive LLM diversity-enhancing generation methods, such as sampling decoding with varying temperatures and diverse beam-search (Vijayakumar et al., 2016). We’ll make this clearer in the revised version.
> >
> > **Markovian Flow vs MCTS**
> >
> > We’d like to clarify that MCTS is not Markovian Flow. While both have the common “Marokovian” assumption, MCTS, as a reinforcement learning approach, aims to find the maximum-reward solution (via Monte Carlo search). In contrast, Markovian Flow builds on the classical concept of “flow”, which enables sampling proportional to an unnormalized reward.
> >
> > We’ve extensively cited/discussed prior works on Markovian Flow in our paper (Introduction, Related Work sections). One of our key technical contributions is the new formulation that applies the classical Markovian Flow perspective to the new setting of multi-step LLM reasoning, which enables LLMs to generate diverse reasoning trajectories. We’ve also discussed the difference from MCTS in several places (such as Line.136) and empirically compared with MCTS method like RAP extensively (such as in Tables.1, 2, and 5).

---

### Meta-Review · Area_Chair_EHDy · 2024-12-20

**Metareview:**

The paper presents Flow of Reasoning (FoR), a fine-tuning approach for LLMs that aims to enhance solution diversity. FoR frames the reasoning process as a Markovian flow, sampling diverse reasoning paths with probabilities aligned to the reward associated with each target problem similar to GFlowNet. Extensive experiments across multiple domains and baselines demonstrate that the proposed approach helps LLMs discover more diverse and higher-quality answers, showcasing the effectiveness of the algorithm.

On the positive side, the paper approach appears to be novel and new. However, some of the writing is not clear, and additional baselines would be helpful. In addition, when I carefully read the paper, the authors make a big deal above sampling trajectories according to the boltzmann distribution over rewards. However -- this is the exact density that is used in KL regularized RL such as DPO and PPO so it would be helpful to clarify in the paper how flow of reasoning is different than other existing approaches.

**Additional Comments On Reviewer Discussion:**

There was extensive discussion during the reviewer discussion period. Unfortunately, 3 reviewers were not convinced about the paper, citing issues with clarity as well as baselines.

---

### Decision · Program_Chairs · 2025-01-22

Reject